# Regulation of the COPII secretory machinery via focal adhesions and extracellular matrix signaling

Juan Jung[1]*, Muzamil Majid Khan[1,3]*, Jonathan Landry[2], Aliaksandr Halavatyi[2], Pedro Machado[2], Miriam Reiss[1], and Rainer Pepperkok[1,2,3]

Proteins that enter the secretory pathway are transported from their place of synthesis in the endoplasmic reticulum to the Golgi complex by COPII-coated carriers. The networks of proteins that regulate these components in response to extracellular cues have remained largely elusive. Using high-throughput microscopy, we comprehensively screened 378 cytoskeleton-associated and related proteins for their functional interaction with the coat protein complex II (COPII) components SEC23A and SEC23B. Among these, we identified a group of proteins associated with focal adhesions (FERMT2, MACF1, MAPK8IP2, NGEF, PIK3CA, and ROCK1) that led to the downregulation of SEC23A when depleted by siRNA. Changes in focal adhesions induced by plating cells on ECM also led to the downregulation of SEC23A and decreases in VSVG transport from ER to Golgi. Both the expression of SEC23A and the transport defect could be rescued by treatment with a focal adhesion kinase inhibitor. Altogether, our results identify a network of cytoskeleton-associated proteins connecting focal adhesions and ECM-related signaling with the gene expression of the COPII secretory machinery and trafficking.

## Introduction

The secretory pathway is responsible for the delivery of proteins and lipids to their correct location, including the plasma membrane, extracellular space, and many membrane-bounded organelles. This allows the cells to dynamically regulate the size, composition, and function of those compartments. Alterations in the secretory pathway play important roles in many diseases including cancer, fibrosis, and neurodegenerative disorders (for reviews see De Matteis and Luini [2011]; Yarwood et al. [2020]), and understanding how cells integrate different internal and external stimuli to fine-tune secretion remains a central question in cell biology and disease mechanisms.

In the early secretory pathway, proteins are transported from the ER to the Golgi complex in a process mediated by coat protein complex II (COPII)–coated carriers. COPII assembly is initiated by the binding of the GTPase SAR1 to specific sites of the ER termed ER exit sites (ERES), followed by the recruitment of an inner coat composed of SEC23-SEC24 dimer responsible for cargo sorting and an outer coat composed of SEC13-SEC31 that provides a structural scaffold for membrane deformation and carrier formation (see Miller and Schekman [2013] and references therein).

At ERES, protein secretion is regulated through an expanded COPII protein interaction network that ensures the secretion meets physiological demands. For instance, nutrient deprivation reduces secretion by decreasing the number of ERES (Zacharogianni et al., 2011). Conversely, growth factor signaling increases the number of ERES and prepares the cells for a higher secretory demand (Farhan et al., 2010; Tillmann et al., 2015). More recently, EGF receptor (EGFR) signaling has also been linked to the transcriptional regulation of the COPII machinery in response to extended EGFR degradation to restore efficiently the availability of EGFR at the plasma membrane (Scharaw et al., 2016).

The cytoskeleton plays an important role in the organization and function of the secretory pathway by acting as a scaffold for membrane deformation, carrier transport, and organelle positioning (Anitei and Hoflack, 2011; Gurel et al., 2014; Fourrière et al., 2020). For example, it has been shown that SEC23 interacts with DCTN1 (p150-glued), a subunit of the dynactin complex that links the COPII machinery to the microtubules, facilitating the long-distance movement of carriers toward the Golgi (Watson et al., 2005). A role of this SEC23–DCTN1 interaction in cargo concentration at ERES independent of microtubules has also been proposed (Verissimo et al., 2015). Less explored in the context of the regulation of the secretory pathway is the function of hundreds of cytoskeleton-associated

..................................................................................................................................................................................................................

[1]Cell Biology and Biophysics Unit, European Molecular Biology Laboratory, Heidelberg, Germany;   [2]Core Facilities Unit, European Molecular Biology Laboratory, Heidelberg, Germany;   [3]Translational Lung Research Center Heidelberg, German Center for Lung Research, Heidelberg, Germany.

Correspondence to Juan Jung: juan.jung@embl.de;   Rainer Pepperkok: pepperko@embl.org

*J. Jung and M.M. Khan should be considered joint first authors.   P. Machado's present address is Centre for Ultrastructural Imaging, King's College London, London, UK.



and regulatory proteins that participate in the integration of extracellular and intracellular information (Moujaber and Stochaj 2019) including, for instance, mechanotransduction (Sun et al. 2016), metabolism, and signaling (Janmey 1998). To systematically and comprehensively explore the functional interactions of the cytoskeleton and associated factors with the transport machinery of the early secretory pathway, we conducted a functional siRNA-based interaction screen exploring possible synergistic interactions of 378 cytoskeleton and associated proteins with the COPII component SEC23A or SEC23B.

## Results and discussion

### Identification of cytoskeleton-related proteins functionally interacting with SEC23A/B

Single siRNAs targeting altogether 378 cytoskeleton and associated proteins were cotransfected with siRNAs targeting SEC23A or SEC23B, and biosynthetic transport of the vesicular stomatitis virus G protein (VSVG) from the ER to the cell surface (Kreis and Lodish 1986) was quantified as previously described (Simpson et al., 2012; Fig. 1 A). To identify synergistic functional interactors of SEC23A or SEC23B, the siRNA concentrations and transfection conditions were chosen such that transfection of SEC23A- or SEC23B-targeting siRNAs alone did not significantly affect VSVG transport compared with control transfected cells (Fig. 1 B). However, cotransfection of siRNAs targeting SEC23A and SEC23B under these conditions caused significant transport inhibition of VSVG at the ER level (Figs. 1 B and 1 E) consistent with their role in cargo sorting and concentration at ERES.

Analysis of more than 85,000 images allowed us to rank the proteins targeted by the respective siRNAs according to their effect on VSVG transport when co–knocked down with either SEC23A or SEC23B (Table S1). Comparing transport scores of double knockdowns to single knockdowns and ranking of the strongest transport effectors revealed 20 cytoskeleton-related genes in which VSVG transport in the double knockdown (SEC23A or SEC23B + cytoskeleton-associated protein) was significantly different from the additive effect one would expect from the results of the single knockdowns (Fig. 1 B).

The synergistic behavior in transport induced by the siRNA depletion strongly suggests a functional interaction between SEC23 and these 20 cytoskeleton-related proteins selected by ranking; henceforth we call these proteins SEC23 functional interactors. We found SEC23 functional interactors that, in the double knockdown, led to transport acceleration, while others led to transport inhibition. Interestingly, no overlap between SEC23A and SEC23B interactors was observed (Fig. 1 B and Table S1). The full results of the screen are available in Table S2.

Annotation of the subcellular location using existing information in Compartments (Binder et al., 2014) and the Human Protein Atlas (Thul et al., 2017) shows that many of the interactors are proteins located at the plasma membrane with a few also localizing to the cytosol, nucleus, or Golgi complex (Fig. 1 C). Interestingly, none of the SEC23 functional interactors were annotated to localize to the ER or ERES as their main locations, which is the place of function for the COPII complex. Previous works have demonstrated a significant role of microtubules and

associated proteins in regulating early secretory pathway (Scales et al. 1997; Presley et al., 1997; Watson et al., 2005; Brown et al. 2014; Verissimo et al., 2015). Therefore, we expected to find in our screen many proteins with a more canonical role in the cytoskeleton, such as motor or alike proteins. While some of the interactors are motor proteins (KIF9, MYO16, KIF3B) or proteins that participate in the organization of the cytoskeleton (MACF1), most of the interactors are proteins involved in signaling pathways (Fig. 1 B) that regulate the cytoskeleton in response to diverse stimuli including growth factor, actin signaling, and cell–matrix interactions (Fig. 1 D). The majority of these 20 proteins have not been previously implicated in the secretory pathway. Furthermore, none of them were identified as hits in an earlier genome-wide siRNA screen that identified several hundred human genes directly or indirectly involved in biosynthetic transport from the ER to the plasma membrane (Simpson et al., 2012) or in another genomewide screen in *Drosophila* (Bard et al., 2006). This demonstrates the potential of such double-knockdown functional interaction screens as a tool to identify new regulators of different cellular processes including, for instance, cholesterol metabolism (Zimoń et al., 2021), cancer (Laufer et al., 2013), and chromatin remodeling (Roguev et al., 2013).

### Functional interactions at the ER exit level

Because the transport score used to select the hits in the screen does not allow us to distinguish the individual intermediate steps in the transport of VSVG from the ER to the plasma membrane, we assessed the intracellular location of VSVG in the double-knockdown combinations causing transport inhibition, as the place of VSVG inhibition in the secretory pathway is expected to give first insights into the mechanism of how the respective target genes may functionally interact. VSVG transport inhibition was observed when SEC23B siRNA was combined with siRNAs targeting any of these seven proteins: CRKL, FERMT2, MACF1 MAPK8IP2, NGEF, PIK3CA, and ROCK1. For six of them (except for CRKL), VSVG was largely retained in the ER, similar to double knockdown of SEC23A and SEC23B, while in control transfected cells, VSVG was observed mainly at the plasma membrane (Fig. 1 E). Only for the double knockdown of SEC23B and CRKL, post-ER structures could be observed (Fig. 1 E).

To gain more insight into the functional interactions, we focused our analyses on MACF1, as this protein has been previously also implicated in the secretory pathway. MACF1 is a microtubule-actin cross-linker (Leung et al., 1999; Karakesisoglou et al. 2000) localized at focal adhesions (FAs) and cell periphery but also present at the Golgi complex (Lin et al., 2005), where it has been proposed to be involved in Golgi-complex-to-plasma-membrane transport (Kakinuma et al., 2004; Burgo et al., 2012). To characterize the MACF1/SEC23B knockdown-induced transport impairment in more detail, the VSVG transport marker was released from the ER for 5 and 10 min only (Fig. 2 A). In control cells, transfected with non-targeting siRNA, VSVG appeared in post-ER punctate structures visible already 5 min after temperature shift, and a substantial amount of VSVG accumulated in the perinuclear region at 10 min (Fig. 2 A). In the case of MACF1/

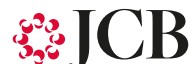

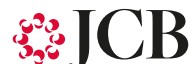

Figure 1. **A functional interaction screen between SEC23 and cytoskeleton-associated proteins uncovers new proteins connected to secretion.**
**(A)** Schematic representation of the high-throughput screen workflow. HeLa cells were cotransfected with siRNAs targeting 378 cytoskeleton-associated proteins and siRNAs targeting the COPII subunits SEC23A or SEC23B. Using an automated microscopy acquisition and analysis pipeline, VSVG transport to the cell surface was quantified after 60 h of depletion in single- and double-knockdown (k.d.) conditions. For each condition, a transport score was calculated. **(B)** Hit selection. The conditions in which the double-knockdown score was significantly different from the expected additive effect of the single knockdowns were selected. Gray circles denote single knockdowns. Red circles denote the positive controls. Interactors in double knockdown are shown in orange (SEC23A interactors) or blue (SEC23B interactors). **(C)** Annotation of the subcellular location of the interactors using the Human Protein Atlas and Compartments as the main sources. For proteins with multiple locations, only the two main locations were annotated (plasma membrane [PM]). Others include mitochondria, centrioles, FAs, etc. **(D)** STRING network (Szklarczyk et al., 2019) using medium confidence and removing the text mining. Proteins in gray are not interactors

but were included to fill up the gaps in the network. **(E)** VSVG transport assay. Wide-field images of the VSVG-YFP after 1 h of temperature shift from 40° to 32°C. Transport was assessed after 60 h of knockdown with the indicated siRNAs. Arrowhead indicates plasma membrane. Arrows indicate ER membranes. Asterisk indicates Golgi or post-Golgi membranes.

SEC23B double-knockdown cells, occasional post-ER punctate structures became visible only 10 min after the temperature shift, and accumulation of the marker in the perinuclear area apparently did not occur (Fig. 2 A). Consistent with this transport impairment at the ER level upon MACF1/SEC23B double knockdown, we observed a reduction in the number of SEC31A-positive ERES (Fig. 2, B and D) and a reduction of COPI-positive ER-to-Golgi-complex transport intermediates (Fig. 2, C and E) compared with control transfected cells or cells transfected with the respective siRNAs alone. Moreover, P24, a cargo receptor that recycles between the Golgi complex and the ER, shifts its distribution to a more ER-predominant localization (Fig. S1 A). In MACF1/SEC23B double-knockdown cells, the Golgi marker GM130 appeared more dispersed compared with control transfected cells, but no striking differences were observed (Fig. S1 B). Also, no apparent changes were observed for the endosomal marker EEA1 (Fig. S1 C). To confirm independently the transport inhibition observed for VSVG, we quantified the transport of E-cadherin to the plasma membrane using the established Retention Using Selective Hooks (RUSH) inducible transport system (Boncompain et al., 2012; Fig. 2, F and G). MACF1/SEC23B double knockdown showed a reduction of E-cadherin transport of 40–70%, depending on the siRNA used, while the single knockdowns did not show a significant transport alteration (Fig. 2 F). In agreement with the data obtained for VSVG, the majority of E-cadherin was retained in the ER at time points when in control transfected cells the transport marker already accumulated in post-ER structures resembling most likely the Golgi complex (Fig. 2 G). Moreover, in the absence of any overexpressed cargo, EM analyses showed that in MACF1/SEC23B double-knockdown cells, the ER becomes often bloated, characteristic of a transport block at the ER level (see, e.g., Fujiwara et al., 1988; Zhang et al., 1994; Fig. S1, D and E). While MACF1 depletion might have effects on other transport steps (i.e., Golgi to plasma membrane, see above), it is clear from our assays that in MACF1/SEC23B double knockdown, the cargo marker does not progress significantly past the ER (Figs. 1 E and 2 A).

Next, we asked whether the functional interaction between MACF1 and SEC23B is due to a physical interaction. While most of the MACF1 staining is located in the cytosol, cell periphery, and Golgi (as previously described in the literature), with no significant staining in the ER or peripheral ERES, we cannot discard colocalization with ERES in the densely populated perinuclear region (Fig. S1 F). To test if MACF1 and SEC23B interact physically, we performed coimmunoprecipitation (co-IP) experiments but failed to detect an interaction under our experimental conditions (Fig. S1 G). While these results do not exclude completely the possibility of physical interaction of SEC23B and MACF1, they suggest that the functional interaction of these two proteins as observed here is most likely indirect.

## A link between cytoskeleton-related proteins and the expression of SEC23A

The striking similarity of the VSVG transport block phenotype for six of seven SEC23B functional interactors in the double-knockdown experiments suggests that there might be a common mechanism or pathway by which these proteins functionally interact with SEC23B. Based on their localization and functional annotation reported in the literature, we considered them unlikely to be direct physical interactors of SEC23B, although this possibility cannot be rigorously excluded. We speculated that their knockdown in cells may affect factors that are more directly related to the membrane traffic machinery in the early secretory pathway. As a first step to investigate this hypothesis, we performed mRNA sequencing to explore the transcriptional changes induced by depletion of these seven interactors alone and test if their depletion might cause overlapping transcriptional changes. We found significant changes in gene expression for all the knockdowns tested (Fig. S2 A and Table S3), and of the ~2,300 unique genes differentially expressed across all seven conditions tested, we observed little overlap between the transcriptional profiles (Fig. S2 B). As expected, depletion of a protein with pleiotropic functions such as PIK3CA generated the biggest changes (i.e., 419 genes downregulated and 580 genes upregulated).

In a more focused approach, we analyzed a set of 595 secretory pathway–related genes based on a recently annotated gene list (Feizi et al., 2017). Overall, the transcriptional changes observed for those genes were relatively minor, with no particular enrichment of these genes or parts of them in any condition (Fisher's exact test). Surprisingly, we found that for all six SEC23B interactors that showed strong ER cargo retention (see Fig. 1 E), *SEC23A* mRNA was significantly downregulated, and was the only common gene that was affected by knockdown (Fig. 3 A and Table S4). The downregulation of *SEC23A* upon knockdown of the interactors could also be confirmed independently by quantitative RT-PCR (RT-qPCR) in HeLa cells and also in human lung fibroblasts for the case of MACF1 siRNA (Fig. 3 B). The decrease in *SEC23A* mRNA levels is also reflected at the protein level (Fig. 3 C). Importantly, the transport phenotype in double-knockdown cells can be rescued by restoring the levels of SEC23A with the overexpression of SEC23A-YFP (Fig. 3 D). These results offer a straightforward explanation for the ER block in the double-knockdown experiments described here, namely, reduction in the levels of total SEC23 by direct downregulation of SEC23B and an indirect downregulation of SEC23A (Fig. S2 C). Notably, while the rescue for FERMT2/SEC23B double knockdown is complete, the rescue for MACF1/SEC23B is not (Fig. 3 D). To explain this, we hypothesize that the overexpression of SEC23A-YFP restores ER-to-Golgi transport but does not rescue the additional roles of MACF1 in the secretory pathway, for instance in this case, transport from the Golgi to the plasma membrane (Kakinuma et al., 2004; Burgo et al., 2012). In light of recent evidence that shows the tubular nature



Figure 2. **MACF1 and SEC23B codepletion impairs ER to Golgi transport. (A)** VSVG transport assay. Confocal images of VSVG-YFP at $t$ = 0 (40°C) and after 5 and 10 min of temperature shift from 40° to 32°C. Transport was assessed after 60 h of knockdown with the indicated siRNAs. Arrows indicate puncta most likely representing transport intermediates. Arrowhead indicates juxtanuclear structures most likely representing Golgi. **(B and C)** Confocal images of ERES assessed by SEC31A staining (B) and transport intermediates assessed by β-COPI staining (C) after 48 h of knockdown with the indicated siRNAs. **(D and E)** Quantification of a representative experiment shown in B and C, respectively. Dots represent individual cell values. Lines represent the mean number of structures per cell. **(F)** E-cadherin RUSH transport assay after 48 h of knockdown with the indicated siRNAs. The bars represent the transport mean of one representative experiment in which >500 cells per condition were quantified and normalized to control siRNA. **(G)** Wide-field images of RUSH E-cadherin–GFP at $t$ = 0 (untreated) and after 45 min of biotin release. Arrows indicate ER membranes. Arrowheads indicate post-ER structures. Statistical significance: ***, P < 0.001 compared with control. ###, P < 0.001 compared with SEC23B or MACF1 single knockdowns, Student's $t$ test.

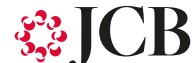

**Figure 3.** **The transport defect phenotype is caused by the downregulation of SEC23A. (A)** Differential expression analysis of a subset of 595 manually curated secretory pathway genes for the indicated knockdowns (48 h) compared with control siRNA. The zoomed-in region highlights the COPII genes. **(B)** RT-qPCR quantification of *SEC23A* levels after 48 h of knockdown with the indicated siRNAs. Bars represent averages of three independent experiments, and dots, the individual values. The dark gray bar represents *SEC23A* levels after 48 h of MACF1 siRNA treatment in human lung fibroblasts. **(C)** Western blot analysis of SEC23A protein levels after 48 h of depletion with the indicated siRNAs. α-Tubulin was used as a loading control. **(D)** SEC23A rescue experiment. Cells were cotransfected with the indicated siRNAs and a cDNA encoding SEC23A-YFP. After 60 h of transfection, the VSVG assay was performed. Based on the intensity of the YFP channel, cells were divided into non-expressing, low-expressing, and high-expressing. Bars represent averages of three independent experiments, and dots, the individual values. Statistical significance: *, P < 0.05 and **, P < 0.01 compared with control, Student's *t* test. Source data are available for this figure: SourceData F3.

of ERES and their relationship to transport intermediates (Weigel et al., 2021) it should be also considered that MACF1 might play a role in ER-to-Golgi transport due to its actin-microtubule crosslinking role (Leung et al., 1999; Karakesisoglou et al. 2000). Indeed, it has been shown that microtubules and actin participate in the tubulation and elongation of membranes moving from the ER to Golgi (Campellone et al., 2008), so the depletion of MACF1 could affect the stability and morphology of ERES or transport intermediate membranes via its cytoskeletal properties and cause a partial transport delay independently of the expression of SEC23A.

### ECM-related cues are connected to SEC23A expression and affect trafficking

The decrease in SEC23A induced by the depletion of particular cytoskeleton regulatory and associated proteins suggests that cytoskeleton-related cues are connected to SEC23A gene expression. But which are these cues?

Enrichment analysis on the mRNA sequencing (mRNA-seq) data showed that the most common process changed among the analyzed proteins was cell adhesion (Fig. 4 A and Table S5). In fact, four out of these six functional interactors connected to the expression of SEC23A are bona fide members of the adhesome (Zaidel-Bar et al., 2007), which currently includes 232 proteins known to be an integral part of FAs or associated with them (www.adhesome.org; Fig. S3 A). Consistent with the idea that knockdown of these proteins induces changes in cell adhesion, we observed morphological alterations in FAs as assessed by vinculin staining for all the knockdowns (Fig. 4 B). For instance, for all the conditions tested, we observed enlarged FAs alongside an increase in the total intensity of FA-associated vinculin per cell. FA number was increased for most of the conditions except

**A**

**Biological processes**

*Actin cytoskeleton organization*
*Anion transport*
*Apoptotic signaling pathway*
*Aromatic compound catabolic process*
*Blood vessel development*
*Carbohydrate derivative biosynthetic process*
*Chemotaxis*
*Extracellular structure organization*
*Lymphocyte activation*
*Monocarboxylic acid metabolic process*
*Muscle structure development*
*Negative regulation of cell differentiation*
*Negative regulation of cell proliferation*
*Nucleoside phosphate metabolic process*
*Organelle localization*
*Positive regulation of locomotion*
*Regulated exocytosis*
*Regulation of body fluid levels*
*Regulation of cell adhesion*
*Regulation of cellular response to stress*
*Regulation of growth*
*Reg. of protein serine/threonine kinase activity*
*Response to growth factor*
*Response to nutrient levels*
*Response to virus*
*Response to wounding*
*TM receptor protein tyrosine kinase signaling*

Enrichment   ○ >1.5   ○ >2.5   ○ >3.5
-Log10 p value   🟡 >2   🟠 >5   🔴 >8

siFERMT2   siMACF1   siMAPK8IP2   siNGEF   siPIK3CA   siROCK1

**B**

Vinculin

siCONTROL | Zoom
10 µm | 5 µm
siMACF1 | Zoom
siNGEF | Zoom
siROCK1 | Zoom
siFERMT2 | Zoom
siMAPK8IP2 | Zoom
siPIK3CA | Zoom

**C**

FA   Total intensity (A.U)   Number per cell   Size (number of pixels)

siControl
siFERMT2
siMACF1
siMAPK8IP2
siNGEF
siPIK3CA
siROCK1

⚪ n.s   🟠 p<0.001   🔴 p<0.0001

Figure 4.   **Knockdown of the SEC23 interactors causes changes in FAs. (A)** Gene enrichment analysis was done using Metascape. GO terms for biological processes with a minimum enrichment of 1.5 and a minimum P value of 0.01 are depicted. When possible, child terms were used over parents, and redundant pathways were omitted. The most common processes across all six conditions are underlined. **(B)** Confocal images of FAs assessed by vinculin staining after

48 h of treatment with the indicated siRNAs. Arrows show the FAs in control cells. Arrowheads show enlarged FAs in knockdown cells. **(C)** Quantification of the experiment shown in B. Dots represent mean values per cell. Lines represent the mean values per condition. Colors represent statistical significance calculated using Student's *t* test.

for FERMT2 and MACF1 knockdowns (Fig. 4 C). Some of these changes are in agreement with what has been described previously, for instance, the enlarged FAs observed in MACF1 knockdown cells (Wu et al. 2008), while some discrepancies with the literature might be attributed to the particular cell type used. In particular, depletion of FERMT2 in fibroblasts leads to a rounded morphology and almost absence of FA (Theodosiou et al., 2016), whereas in podocytes it leads to fewer FAs but an increase in their size (Qu et al., 2011). In our knockdown experiments, HeLa Kyoto cells remained well attached and mostly spread, and we never observed a significant reduction in FA number (Fig. 4 C).

Changes in FAs are characteristic of changes in adhesion observed when cells interact with different substrates (see Fig. S3 B), and one of the main transducers of these changes is the FA-associated kinase (FAK; see review by Geiger et al. [2009]). To directly test if cell adhesion and FAK signaling are involved in the regulation of SEC23A, we first treated cells with the FAK inhibitor PND-1186. As seen in Fig. 5 A, PND-1186 treatment upregulated specifically *SEC23A* but not *SEC23B* expression. Of note, PND-1186 treatment could only partially counteract the decrease in *SEC23A* levels induced by depletion of FERMT2 or MACF1 (Fig. S3 C). Second, HeLa cells plated on human primary fibroblast–derived ECM showed reduced *SEC23A* mRNA levels compared with cells plated on plastic alone, whereas *SEC23B* remained unchanged (Fig. 5 B). Plating the cells on Matrigel induced a similar decrease in *SEC23A* (Fig. 5 B), and given the availability and defined composition of Matrigel, we decided to use it as a substrate for our next experiments. Interestingly, the Matrigel-mediated downregulation of *SEC23A* could be prevented completely by treatment with PND-1186 (Fig. 5 B). As is the case for the knockdowns, the Matrigel-induced decrease in *SEC23A* mRNA was also observed at the protein level with no apparent difference between the different Matrigel concentrations (Fig. 5, C and D). Of note, the decrease in SEC23A protein level in Matrigel appears to be less dramatic than the decrease induced by the depletion of the interactors. This is consistent with the changes in the mRNA levels. Depletion of interactors decreases *SEC23A* mRNA by 50–60% (Fig 3 B), but Matrigel induces only a 25–40% decrease in *SEC23A* mRNA (Fig. 5 B).

Finally, we tested if the SEC23A downregulation induced by adhesion to Matrigel has an effect on trafficking. To set up this experiment, we first looked for the conditions in which a decrease in SEC23A would affect VSVG trafficking. Here we observed that in low- to medium-level VSVG-expressing cells, depletion of SEC23A has a low impact on VSVG trafficking (~10% decrease), most likely because SEC23B is enough to compensate for the lack of SEC23A. This was the predominant condition used during the screen, as most of the cells express these levels of VSVG. The decrease in SEC23A becomes more relevant as cells express higher cargo levels (~30% decrease in

VSVG transport; Fig. S3 D). To analyze the effect of Matrigel in trafficking, we followed a similar approach, dividing the cells according to the VSVG level of expression. In low- to medium-level VSVG-expressing cells, plating the cells on Matrigel generates a minor effect on transport, while for high-level expressing cells, the effect is more evident (30.5% [0.4 µg/ml] and 19.9% [4 µg/ml]) decrease in VSVG transport; Fig. 5 E). Of note, the decrease in VSVG transport at 0.4 and 4 µg/ml of Matrigel was not significantly different (Fig. 5 E), in agreement with changes in SEC23A expression at these concentrations of Matrigel (Fig. 5 C). Similarly to what happens with the expression of SEC23A, the decrease in transport induced by Matrigel could be counteracted by PND-1186 (Fig. 5 E). Therefore, this type of regulation becomes relevant when cells have to secrete a large amount of cargo or when cells have to secrete cargoes that depend mainly on SEC23A, as has been proposed for collagens (Boyadjiev et al., 2006; Lang et al., 2006). The ECM-mediated regulation of SEC23A might also be relevant for cancer. It has been shown that downregulation of SEC23A leads to changes in the secretome of cancer cells that promote metastatic capabilities (Korpal et al., 2011; Szczyrba et al., 2011). As metastatic cancer cells migrate and interact dynamically with the ECM, it is plausible that signals arising from the ECM contribute to SEC23A regulation in cancer cells, although this needs further investigation.

FAs have been previously involved in membrane trafficking, including endocytosis and exocytosis (Wickström and Fässler 2011; Nolte et al. 2020). In exocytosis, FAs appear to serve as hotspots for secretion, a function that might be linked to the role of FAs in the organization of microtubules from TGN to plasma membrane (Stehbens et al., 2014; Fourriere et al., 2020; Eisler et al., 2018). Based on our results here, we propose that ECM and FAs are also connected to membrane trafficking through the regulation of SEC23A expression. In the context of low FAs or low FAK signaling (i.e., cells plated on plastic in our experiments), the levels of SEC23A are high. In the context of high FA or FAK signaling (i.e., cells plated on ECM), SEC23A is downregulated, contributing to trafficking regulation in response to cell adhesion (Fig. 5 F). These views are complementary and help to understand how cells can connect trafficking and extracellular cues in a robust manner. The precise molecular cascade that connects ECM and adhesion to the expression of SEC23A and its physiological implications remain to be investigated.

## Materials and methods
### Cell lines and reagents
HeLa Kyoto cells were a gift from Shuh Narumiya (Kyoto University, Kyoto, Japan) and were cultured in DMEM supplemented with 10% FCS and 2 mM glutamine. Normal human primary lung fibroblasts were obtained from Lonza and were cultured in fibroblast growth basal medium (#CC-3131). HeLa

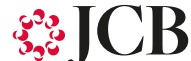

Figure 5. **FAK signaling and cell adhesion to ECM are connected to SEC23A expression. (A)** *SEC23A* and *SEC23B* mRNA levels were assessed by RT-qPCR after treatment with the FAK inhibitor PND-1186 for the indicated times at the indicated concentrations. **(B)** *SEC23A* and *SEC23B* mRNA levels were assessed by RT-qPCR after 48 h of plating cells in dishes coated with fibroblast-derived ECM or Matrigel (4 μg/ml) and treated with the FAK inhibitor PND-1186 for 24 h at the indicated concentrations. **(C)** Western blot analysis of SEC23A protein levels after 72 h of plating cells in dishes covered with the indicated concentrations of Matrigel. **(D)** Quantification of the Western blots. **(E)** VSVF transport experiment. Cells were plated at the indicated concentration of Matrigel, and the VSVG

was performed 72 h later. Some dishes were treated with PND-1186 48 h before the VSVG release. Cells were divided according to the level of expression of VSVG. A minimum of 50 cells per condition were analyzed. **(F)** Model representing the proposed relationship between adhesion, FA signaling, and SEC23A. (1) Interaction of cells with ECM or substrate through FAs. (2) Activation of FAK and a downstream signaling cascade. (3) Transcriptional regulation of *SEC23A*. (4) SEC23A-mediated ER-to-Golgi transport. In a condition of low FAs or low FAK signaling (left), *SEC23A* gene expression is not repressed. SEC23A levels are high (blue circles) and SEC23A-dependent transport proceeds normally (green arrow). In a condition of high FA or high FAK signaling (right) the repression is more prominent, and SEC23A levels decrease, possibly affecting SEC23A-dependent transport. The elements of the signaling cascade and the transcriptional regulation remain to be identified. For the experiments presented in B–D, dots represent values obtained from independent experiments, and the bar, the mean value per condition. For E, dots represent the values per cell, and the lines represent the mean value per condition. Statistical significance: *, $P < 0.05$; **, $P < 0.01$; ***, $P < 0.001$ compared with control; ###, $P < 0.001$ compared with untreated cells, Student's *t* test. In D, n.s., non significant, by one-way ANOVA among different Matrigel concentrations. Source data are available for this figure: SourceData F5.

cells stably expressing P24-YFP (TMED2) were generated in-house and described before (Simpson et al. 2006). All cells were kept in a humidified incubator at 37°C and 5% $CO_2$.

All siRNAs used were from Ambion and are described in Table S6. Anti-VSVG antibody and recombinant adenovirus expressing YFP-tagged and CFP-tagged tsO45G thermosensitive versions of VSVG were a gift from Kai Simmons (Max Planck Institute of Molecular Cell Biology and Genetics, Dresden, Germany). Large-scale adenovirus preparation was done by Vector Biolabs. RUSH construct encoding E-cadherin–GFP was a gift from Frank Perez (Institut Curie, Paris, France). YFP-SEC23A was described elsewhere (Stephens et al., 2000). Anti-MACF1 antibody was a gift from Ronald Liem (Columbia University, New York, NY). The following primary antibodies were used: SEC23A (ab179811 and ab137583) from Abcam, Vinculin (MAB3574) from Merck, EEA1 (610457) and GM130 (610822) from BD Biosciences, α-tubulin (MS581P1) from Thermo Fisher Scientific, SEC23B (GTX82432) from GeneTex, and GFP (11814460001) from Roche. Anti-β COPI rabbit serum was produced in-house. The following secondary antibodies were used: rabbit anti-mouse-HRP (A9044) from Merck, goat anti-rabbit-HRP (ab6721) from Abcam, and Alexa-conjugated antibodies from Molecular Probes. Hoechst dye 33258 was from Invitrogen. SiR-DNA was from Spirochrome. FAK inhibitor PND-1186 was from Tocris Bioscience. Matrigel was from Corning (#356230, growth factor reduced). Cycloheximide was from Calbiochem. Biotin was from Sigma-Aldrich.

### siRNA-based functional interaction screen

The siRNA cytoskeleton library was designed in-house based on gene annotation and literature to include 848 siRNAs targeting 378 cytoskeleton structural proteins, motor proteins, and other cytoskeleton-associated and regulatory proteins. The siRNAs were selected based on the score and number of transcripts targeted. The cytoskeleton siRNAs were spotted on Nunc Lab-Tek chambers at a density of 384 spots per chamber as previously described (Simpson et al., 2012). The whole cytoskeleton library was distributed on four Lab-Tek chambers, which included also positive controls (β and γ *COPI* targeting siRNAs), two different nontargeting siRNAs (Scramble and Neg9 siRNAs), and an *INCENP* targeting siRNA to evaluate the transfection efficiency of the plate based on the number of multinucleated cells (Neumann et al., 2006). Spotted Lab-Teks were dried, stored in humidity-free containers, and used within a month after coating. At the moment of cell plating, cells were transfected either with *SEC23A*, *SEC23B*, or control siRNAs in suspension at a final

concentration of 15 nM using Lipofectamine 2000 and Opti-MEM (Thermo Fisher Scientific) according to the manufacturer's instructions. 1.5 ml of transfected cell suspension containing 100,000 HeLa cells were plated onto each Lab-Tek. 48 h after plating, cells were infected with adenovirus coding for the VSVG-tsO45-YFP protein (henceforth called VSVG-YFP) for 1 h at 37°C, washed twice, and transferred to 40°C for 12 h to accumulate VSVG-YFP in the ER as described (Simpson et al., 2012). Release of the transport marker from the ER was induced by shifting temperature to 32°C in fresh DMEM containing 25 mM Hepes and 50 µg/ml cycloheximide. After 1 h at 32°C, cells were washed twice with PBS and fixed with 3% PFA for 20 min. VSVG at the plasma membrane was detected by incubating cells for 1 h with an anti-VSVG antibody (1:100) recognizing an extracellular epitope of VSVG, followed by 30-min incubation of anti-mouse Alexa Fluor 647 secondary antibody (1:500). Cell nuclei were stained with Hoechst 33258. Finally, cells were washed twice with PBS and covered with PBS for imaging. Images were acquired on an Olympus ScanR microscope using a 20×/0.7-NA objective. The whole screen was carried out in five independent biological replicates.

### Screen data analysis

Images were analyzed using CellProfiler (Carpenter et al., 2006) as previously described (Scharaw et al., 2016), in which individual cells are segmented and the VSVG specific intensity at the plasma membrane (A647 signal) and the total VSVG intensity (YFP signal) are quantified per cell, and a transport ratio (A647 signal/YFP signal) is calculated. To allow comparison of different biological replicas and to correct for plate effects, transport ratios for each siRNA were converted to transport scores according to the following formula:

$$\text{Transport score} = xi - X/MAD,$$

where $xi$ is the average transport ratio per cell for the siRNA of interest and $X$ is determined as the median of 25 (5 × 5 matrix) transport ratios from neighboring siRNA spots surrounding the siRNA spot of interest. MAD is the median absolute deviation in the 5 × 5 matrix. Negative transport scores represent transport inhibition, and positive transport scores represent transport acceleration.

As quality control, we considered only the plates in which our positive controls showed strong transport inhibition and the negative controls had a score very close to 0. Finally, we considered only the siRNAs for which we had at least three biological replicas and we computed the median transport score of these replicas for each siRNA.

To obtain the hits, we set our threshold on a transport score of –1.5 (strong inhibition) based on our positive controls (*COPB1*, *COPG1*, and *SEC23A/B*). By symmetry, only combinations showing scores >1.5 were considered transport accelerators. Then, we compared added median transport scores of the single knockdowns with the real scores of the double knockdowns and selected the ones in which the difference was >1 transport score unit (Table S1 and table legends). Both nontargeting siRNAs (Scramble and Neg9) had transport scores close to 0 (0.08 and –0.16, respectively), so we selected Neg9 as a control siRNA for all our following experiments. Annotation of the subcellular location of the hits was done using existing information in Compartments (Binder et al., 2014) and the Human Protein Atlas (Thul et al., 2017).

## Confocal microscopy and image segmentation

Confocal stacks were acquired using a Leica SP8 system with a 63×/1.4-NA objective, a Nikon A1 with a 40×/1.3-NA objective, and a Zeiss 780 with a 40×/1.1-NA objective. Unless otherwise stated, the images shown represent maximum-intensity Z-projections. Pipelines for quantification of ERES and COPI transport intermediates were implemented as custom Fiji plugins. ERES and COPI structures were segmented and quantified in 3D in the individual cells using functions of the 3D image suite library (Ollion et al., 2013). First, nuclei were segmented in the Hoechst channel. Cell masks were identified by applying a watershed algorithm in the COPI channel using previously segmented nuclei as seeds. For robust segmentation of ERES and COPI-positive structures, 3D median filters were applied for noise suppression and for subtracting local backgrounds in corresponding channels. Seeds for the potential structures were identified by applying a local maximum filter. Individual structures were segmented using the spot segmentation plugin of the 3D image suite including watershed separation of touching structures. When processing the COPI channel, big positive clusters, which presumably correspond to the Golgi complex, were segmented in 3D by thresholding and excluded from spot analysis. The pipeline documented the number of identified structures in each cell, their integrated intensity, and the integrated intensity of the entire cell. For each condition, a minimum of 100 cells were quantified.

For the segmentation and quantification of FAs in HeLa cells, the plane where FAs are located was identified as the optical slice with the largest normalized variance. Segmentation of FAs was performed in 2D by thresholding on the maximum Z-projections of three confocal slices centered on the identified plane to exclude cytoplasmic staining in other planes from the analysis. The total area of FAs in each image was quantified. Nuclei were segmented in the Hoechst channel to normalize the area of FAs to the number of cells. For each condition, a minimum of 50 cells were quantified.

## RUSH assay

For the RUSH assay, HeLa cells were plated in coverslips in 24-well plates at a density of 15,000 cells per well and, 24 h later, transfected with siRNAs using Lipofectamine 2000 and serum-free DMEM. 24 h later, the cells were transfected with the RUSH construct encoding E-cadherin using Lipofectamine LTX and Enhancer in serum-free DMEM (as per manufacturer's instructions). The next day, the RUSH assay was performed as described previously (Boncompain et al., 2012). In brief, the cell medium was changed to fresh DMEM containing 40 µM biotin and 50 µg/ml cycloheximide. After 45 min, cells were washed once with ice-cold PBS and incubated with anti-GFP (1:100) in PBS on ice for 45 min. After incubation, cells were washed twice with PBS and fixed with PFA 3% at room temperature for 20 min. After washing two times with PBS, cells were stained for 30 min with an anti-mouse Alexa Fluor 647 antibody (1:500). Cell nuclei were stained with Hoechst 33258. Finally, coverslips were mounted on glass slides using Mowiol mounting medium, and images were acquired and analyzed with CellProfiler as above. For every condition, a nonrelease control was also included to account for cargo leakiness.

## EM

Cell monolayers were fixed in 2.5% glutaraldehyde in cacodylate buffer, washed with cacodylate, and postfixed in 1% osmium. En bloc staining was done in 1% uranyl acetate. The cells were then dehydrated in ethanol and embedded in Epon resin. All specimen preparation steps up to embedding were done using a PELCO Biowave Pro microwave (Lorentzen et al., 2018). Thin sections of 70 nm were collected in 2 × 1 slot grids and poststained with lead citrate. The imaging was done using the LLP viewer in a JEOL 2200 Plus electron microscope equipped with a Matataki camera. For image analysis, 20 cells per condition were segmented using ORS Dragonfly. To avoid tangentially oriented ER skewing the analysis, only the ER pieces where the lumen and membranes in cross-section were visible were segmented.

From the segmentation, the maximum distance from the ER lumen to the membrane was calculated and exported as a CSV file.

## Co-IP and Western blot analysis

For co-IP experiments, Hela cells were plated on a 10-cm Petri dish and lysed at ~90% confluency using NP-40 lysis buffer (1% NP-40, 150 mM NaCl, and 50 mM Tris/HCl, pH 7.4) supplemented with protease inhibitors. All subsequent steps were carried out at 4°C. Cells were lysed for 30 min and centrifuged at 20,000 *g* for 15 min. 60 µl of protein G–agarose was added to 400 µl lysate in the presence or absence of 2 µg anti-MACF1 antibody and incubated with rotation overnight. Beads were pelleted and washed three times with lysis buffer and then heated at 60°C for 5 min with sample buffer 2× (10% glycerol, 4.5% SDS, 130 mM DDT, 0.005% bromophenol blue, and 80 mM Tris/HCl, pH 6.8). Proteins were separated by SDS-PAGE using a 3–8% Tris-acetate gel and then transferred to polyvinylidene difluoride membranes for Western blotting.

For SEC23A Western blotting experiments, cells were lysed directly with sample buffer 2×. Lysates were treated with Benzonase for 10 min at room temperature and then heated at 90°C for 5 min. Proteins were separated by SDS-PAGE using a 4–12% Bis-Tris gel and transferred to polyvinylidene difluoride membranes as above.

### mRNA-seq

Individually barcoded stranded mRNA-seq libraries were prepared from high-quality total RNA samples (~600 ng/sample) using the Illumina TruSeq RNA Sample Preparation v2 Kit (Illumina) implemented on the liquid handling robot Beckman FXP2. Obtained libraries that passed the quality control step were pooled in equimolar amounts; 1.8 pM solution of this pool was loaded on the Illumina sequencer NextSeq 500 and sequenced unidirectionally, generating ~500 million reads, each 85 bases long.

Single-end sequencing reads were aligned to the human genome (version GRCh38) and the reference gene annotation (release 84, Ensembl) using STAR v2.6.0a (Dobin et al., 2013) with default parameters. Read counts per gene matrices were generated during the alignment step (--quantMode Gene-Counts). Gene counts were then compared between two groups of replicated samples using the DESeq R package (Love et al. 2014). Differentially expressed genes were selected based on their P adjusted value <10% and their $\log_2$ fold-change >0.58 or less than –0.58.

### RT-qPCR

Total RNA was extracted using RNAeasy kit (Qiagen). 500 ng total RNA was subjected to reverse transcription using SuperScript III First-Strand Synthesis Supermix (Invitrogen) according to the manufacturer's instructions. cDNAs obtained this way were diluted 1/10 and used for PCR. The RT-qPCR reaction was performed using the SYBR green detection reagent (Applied Biosystems) in StepOne Real-Time PCR System machines using the StepOne software v2.3 (Applied Biosystems). Primer sequences are described in the supplementary information. Changes in gene expression were calculated with the $2^{-\Delta\Delta CT}$ method (Livak and Schmittgen 2001) using *GAPDH* as the housekeeping gene.

### SEC23A rescue experiment

Cells plated in 24 glass-bottom well plates were cotransfected with siRNAs and SEC23A-YFP using Lipofectamine 2000. After 48 h, cells were infected with VSVG-CFP–coding adenovirus, and the VSVG assay was carried out as previously indicated with the following modifications to minimize fluorescence bleed-through: the external VSVG was detected using a secondary Alexa Fluor 568–conjugated antibody, and the nuclei were stained in the far red using SiR-DNA. The transport ratios were obtained as previously indicated, and cells were separated into three categories based on the intensity in the YFP channel: nonexpressing cells, low expressing cells, and high expressing cells.

### ECM experiments

For preparing ECM derived from lung fibroblasts, cells were seeded at a density of 20,000–25,000 cells/well in 24 glass-bottom well plates. After 18 h cells were switched to DMEM medium with macromolecular crowding (a mixture of Ficol 70 and Ficol 400; Chen et al., 2009). Cells were cultured for 5 d with stimulation (TGFβ1 5 ng/ml, 240-B-010/CF, R&D Systems) and media change on days 1 and 4. On day 5, cells were washed with PBS, and the plates were subjected to decellularization using 1%

NP-40 (Harris et al. 2018). For Matrigel experiments, Matrigel was diluted in DMEM to create a stock solution of 0.2 mg/ml, frozen at –20°C, and then diluted further in DMEM and used to cover plates according to the manufacturer's instructions at the indicated final concentrations in the experiments.

### Online supplemental material

Fig. S1 contains additional data for the ER block phenotype in MACF1/SEC23B double knockdown. Fig. S2 shows the differential expression analysis. Fig. S3 contains additional data for the FA phenotype. Table S1 shows the SEC23 interactors selected by ranking. Table S2 contains the full screen results. Table S3 contains the full differential expression analysis. Table S4 shows the analysis of 595 secretory pathway–related genes. Table S5 shows the gene enrichment results. Table S6 contains the sequences of all the siRNAs used.

## Acknowledgments

We would like to thank the European Molecular Biology Laboratory Advanced Light Microscopy Facility for help with the preparation and analysis of the screen, the European Molecular Biology Laboratory Genomics Core Facility for help with mRNA-Seq sample preparation and analysis, Seetharaman Parashuraman (Institute of Biochemistry and Cell Biology, Naples, Italy) for useful discussions and critical reading of the manuscript, Charlotte Carr for help with the RUSH assay, and members of the Pepperkok Team for fruitful discussions during the development of this project.

J. Jung was funded by a fellowship from National Commission for Scientific and Technological Research, Chile. M.M. Khan was funded by a Federal Ministry of Education and Research grant (German Centre for Lung Research).

The authors declare no competing financial interests.

Author contribution: Conceptualization: J. Jung, R. Pepperkok, M.M. Khan. Investigation: J. Jung, M.M. Khan. Formal Analysis: J. Jung, M.M. Khan, J. Landry, A. Halavatyi, P. Machado. Resources: P. Machado, M. Reiss. Supervision: R. Pepperkok. Writing original draft: J. Jung, R. Pepperkok, M.M. Khan. Writing review and editing: all the authors. Funding acquisition: R. Pepperkok, J. Jung.

Submitted: 15 October 2021

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

**Supplemental material**

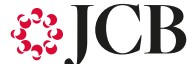



**Figure S1. MACF1 functionally interacts with SEC23B at the ER exit level. (A)** Confocal images of HeLa cells stably expressing P24-YFP and transfected with the indicated siRNAs for 48 h. Arrows indicate ER membranes. **(B and C)** Confocal images of cells transfected with the indicated siRNAs for 48 h, fixed, and stained with anti-GM130 antibody (B) or anti-EEA1 antibody (C). **(D)** EM images of cells treated with the indicated siRNAs for 48 h. Arrows indicate ER membranes. The bloated ER in the MACF1/SEC23B double knockdown is characteristic of an ER transport block. **(E)** Quantification of the EM images. Histogram representing the distribution of ER membranes width. Average (av.) values are also shown. Statistical significance: ***, P < 0.001 compared with control, Student's *t* test; #, P < 0001, $\chi^2$ test for the distribution. **(F)** Confocal images of untreated HeLa cells fixed and costained for MACF1 and SEC31A as a marker for ERES. **(G)** Co-IP experiment. HeLa cells were lysed and incubated with anti-MACF1 antibody bound to agarose-protein beads at 4°C overnight. The beads were washed and loaded into gels. Western blot was performed using anti-MACF1 and anti-SEC23B antibodies. Source data are available for this figure: SourceData FS1.

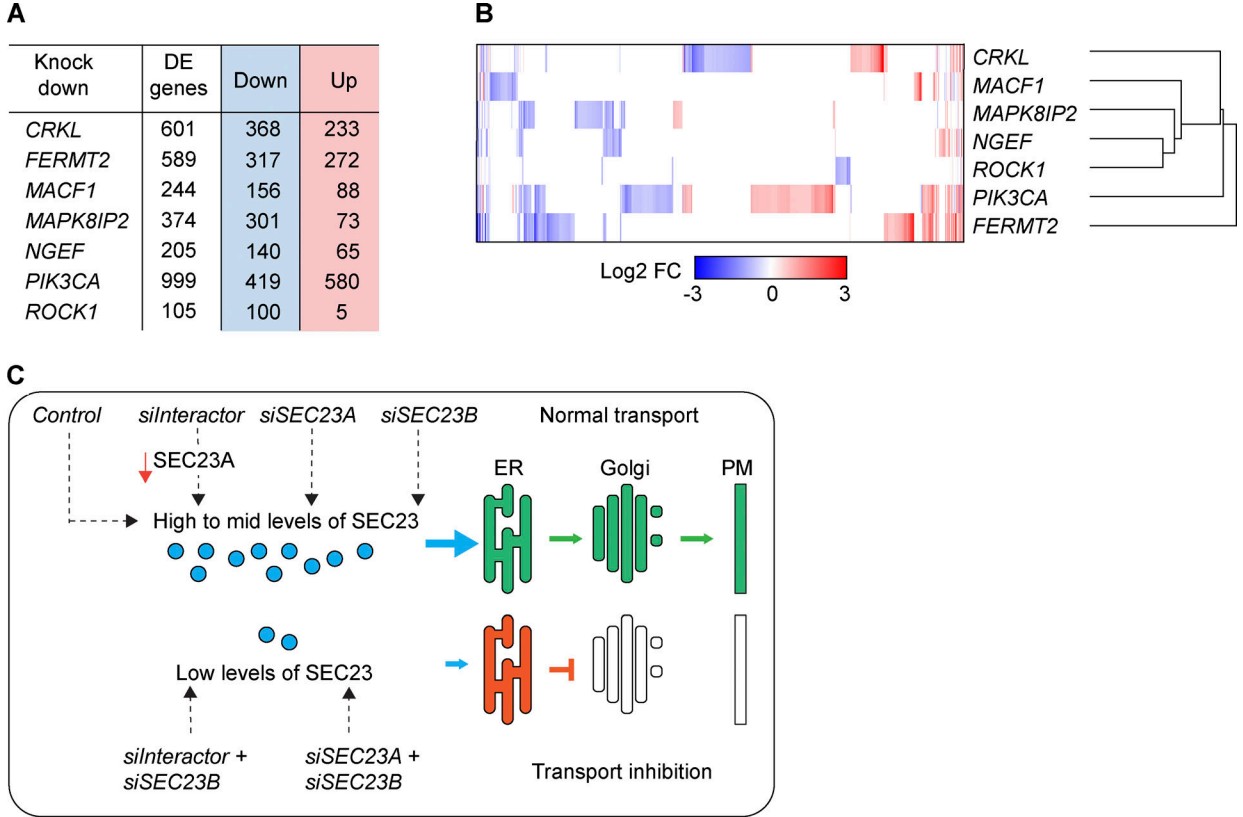

**A**

| Knock down | DE genes | Down | Up |
|---|---|---|---|
| *CRKL* | 601 | 368 | 233 |
| *FERMT2* | 589 | 317 | 272 |
| *MACF1* | 244 | 156 | 88 |
| *MAPK8IP2* | 374 | 301 | 73 |
| *NGEF* | 205 | 140 | 65 |
| *PIK3CA* | 999 | 419 | 580 |
| *ROCK1* | 105 | 100 | 5 |

Figure S2. **Differential expression analysis. (A)** Cells were treated with the indicated siRNAs, and after 48 h, total RNA was extracted, and mRNA was sequenced using an Illumina workflow. The differentially expressed (DE) genes were obtained by comparing expression to control-treated cells with a minimum of three biological replicas per condition. The false discovery rate was set at 0.1, and the $\log_2$ fold-change at 0.58. **(B)** The differential expression analysis was hierarchically clustered and represented as a heatmap using Morpheus (https://software.broadinstitute.org/morpheus). **(C)** Scheme representing the transport phenotypes obtained during the screen and their relationship to SEC23A and SEC23B levels.

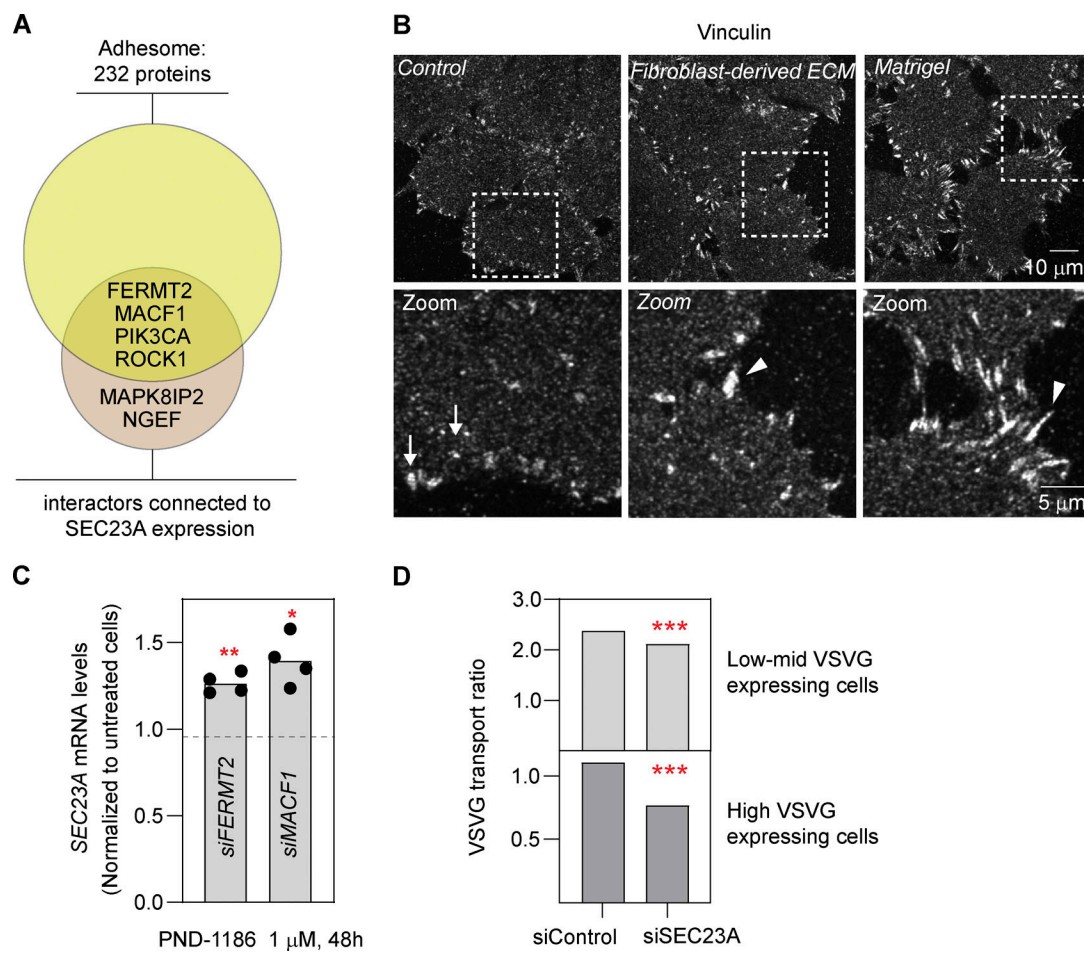

Figure S3. **Additional FA phenotype data and transport experiments. (A)** Venn diagram representing the intersection between the six interactors and the adhesome proteins. **(B)** Confocal images of FAs assessed by vinculin staining after 48 h of the indicated treatment. Arrows show the FAs in cells plated in control (plastic) plates. Arrowheads show the enlarged FAs in cells plated in ECM- or Matrigel-treated plates. **(C)** *SEC23A* mRNA levels were assessed by RT-qPCR in cells treated with the indicated siRNAs together with the FAK inhibitor PND-1186 at the indicated time and concentration. The data were normalized to cells treated with the indicated siRNA but no FAK inhibitor. The dots represent the values obtained for independent experiments, and the bar, the mean value. **(D)** VSVG assay after 48 h of treatment with the indicated siRNAs. Cells were divided according to the level of VSVG expression. Bars represent the average transport ratio per condition of one representative experiment in which ≥100 cells were quantified. Statistical significance: *, $P < 0.05$; **, $P < 0.01$; ***, $P < 0.001$ compared with control, Student's t test.

**Provided online are six tables. Table S1 shows the SEC23 interactors selected by ranking. Table S2 contains the full screen results. Table S3 contains the full differential expression analysis. Table S4 shows the analysis of 595 secretory pathway–related genes. Table S5 shows the gene enrichment results. Table S6 contains the sequences of all the siRNAs used.**

