## [Peer Review File · The Journal of Cell Biology]

Regulation of the COPII secretory machinery via focal adhesions and extracellular matrix signaling

Juan Jung, Muzamil Khan, Jonathan Landry, Aliaksandr Halavatyi, Pedro Machado, Miriam Reiss, and Rainer Pepperkok

Corresponding Author(s): Juan Jung, European Molecular Biology Laboratory and Rainer Pepperkok,

Review Timeline:

Submission Date:	2021-10-15
Editorial Decision:	2021-10-18
Revision Received:	2022-05-10
Editorial Decision:	2022-06-03
Revision Received:	2022-06-23

Monitoring Editor: Elizabeth Miller

Scientific Editor: Andrea Marat

Transaction Report:

DOI: <https://doi.org/10.1083/jcb.202110081>

Revision 0

Review #1

1. How much time do you estimate the authors will need to complete the suggested revisions:

Estimated time to Complete Revisions (Required)

(Decision Recommendation)

Between 1 and 3 months

2. Evidence, reproducibility and clarity:

Evidence, reproducibility and clarity (Required)

The manuscript by Jung et al reports on an interesting finding that focal adhesion signaling regulates the expression of Sec23A and thereby regulates COPII-dependent trafficking. The data presented a mostly solid and the finding itself is highly novel, as it tackles an area of secretory trafficking that remains poorly understood, namely the connection between the ECM and secretion.

I will list below all comments that I have mixing both technical and conceptual topics:

****Technical issues:****

1-The authors should provide a better description of how they designed this siRNA library. What were the inclusion criteria for these 378 genes? I might have missed it, but I could not find this information easily.

2-Figure 2: I know this is challenging for EM images, but is there a way the authors could quantify these data? How many images were looked at? What was the average width of ER cisternae?

3-Figure 4: I think that the characterization of the FA phenotype is a bit underdeveloped. There is no quantification of these data. Is the size of FA changing? Is the number of FA per cell changing? Is the length of FAs changing? I think that more work is needed to increase the confidence in these data.

I could also not easily see what type of cells these are. A better description of this experiment is also required. Also, how many cells were analyzed. I think it is important that this experiment is done with a sufficient number of cells to increase the confidence in the data.

****Conceptual issues:****

1-The finding that focal adhesion signaling negatively affects ER-export is surprising, because cancer cells that grow on stiff substrates have more focal adhesions and are more invasive and migratory. Both migration and invasion are expected to depend on ER-export. Although the authors did not formally test Sec23A expression under different stiffnesses, I would expect that stiff substrates would lower Sec23A expression and thereby negatively affect ER-export. It would certainly increase the breadth of this work to include data like this and to also discuss this highly surprising finding. However, it is of course the decision of the authors and the editors to decide whether such an experiment would benefit the entire story.

2-The authors postulate that this novel mechanism could be part of a feedback loop. If this were the case one would expect the acute effect of FA to increase ER-export (or secretion) and the negative feedback will then reduce secretion. However, the acute effect of FA is not addressed in this manuscript. In order to postulate a feedback loop, the authors would need to test the individual nodes of this loop.

3. Significance:

Significance (Required)

I think that the basic finding of this manuscript is highly novel, by showing the impact of the ECM and focal adhesions on COPII-dependent trafficking. I think that this will not only appeal to people from the trafficking community, but also to people working on cell migration and on mechanobiology. The work in its current form does not require much extra efforts (max. 3 month). However, if the authors would decide to increase the breadth of data, they would require 3-6 months.

****Referees cross-commenting****

I went through the comments of the two other reviewers and agree with their verdict. Some extra work on the characterization of the early secretory pathway would be good. Both reviewers provided a nice catalogue of possible experiments to choose from.

Review #2

1. How much time do you estimate the authors will need to complete the suggested revisions:

Estimated time to Complete Revisions (Required)

(Decision Recommendation)

Between 1 and 3 months

2. Evidence, reproducibility and clarity:

Evidence, reproducibility and clarity (Required)

The manuscript by Jung et al which based on a targeted siRNA screen, demonstrates regulation of SEC23A (component of the SEC23 complex of the COP coat) levels at transcriptional level downstream of focal adhesion signaling. By regulating siRNA mediated downregulation, the authors were able to identify proteins which either increased or decreased traffic of VSVG through the secretory pathway when combined with downregulation in the levels of with either SEC23A or SEC23B. Authors have focused on a group of SEC23B functional interactors, downregulation of which shows them increased size of focal adhesions which also downregulate SEC23A levels, thus providing an explanation for reduced secretory traffic. Authors further show that plating cells on fibronectin or Matrigel, which activate Focal adhesion kinase signaling also results in downregulation of SEC23A transcript levels. The screen is conducted in a well-controlled manner for most parts with a clear explanation of the analysis routines and the data presentation if of very good quality. Most important results have been validated by more than one experimental strategy which lends substantial confidence to the findings. The results also open further avenues for understanding the transcriptional regulation in different physiological and disease contexts.

There are certain issues, which the authors should address with regards to controls and some conflicting observations with published results with respect the phenotypes associated with downregulating proteins on focal adhesions size. Additionally, authors don't tie the ends by monitoring secretory traffic in cells grown on different matrices but include it in the model. Addressing/explaining these issues could improve this manuscript and the model may have to be tweaked a bit.

****Major comments:****

1)I wonder why the authors only used siRNA control in their screen when the effects are scored in context of double knockdown fashion in combination with mild knockdown of SEC23A and SEC23B to get functional interactors. Control siRNA in combination with SEC23A and SEC23B should have been two ideal negative controls in the screen. Nevertheless, in data presented Figure 1E and whole of Figure 2, using control siRNA in combination with SEC23B siRNA would have been ideal control to show that the combination does not induce any trafficking defects which could impact the findings of the study. Hence, a few of the data presented from some of these figures should have sicontrol+SEC23B siRNA combination as a control.

2)What is the identity of post-ER structures which authors refer to in Figure 2A? Could the images represent VSVG concentrated at ER exit sites? Authors should stain with markers for ERES to see if the VSVG puncta colocalize with it.

3)Based on RNA sequencing results, authors chose to follow up on SEC23A levels in background of siRNA knockdown of components (like MACF1, ROCK1, FERMT2 etc.) which regulate Focal adhesions in cells and show that there is a reduction in both transcript and protein levels of SEC23A. In images shown in Figure 2B and Figure 2C, levels of SEC31A and β -Cop1 are reduced. Authors should test using qPCR and western blots whether there is a downregulation of SEC31A, β -Cop1 and SEC23B in siRNA knockdowns of MACF1, ROCK1, FERMT2 etc. It would provide new insights if there were a co-regulation of secretory machinery to modulate the secretory traffic in response to Focal Adhesion based signaling.

4)Most major concern in this manuscript surrounds around results presented in Figure 4C. Authors show that in response to all the knockdowns, they see more focal adhesions as monitored by Vinculin staining and this along with the experiments with cells plated on Matrigel and Fibronectin arrive at the conclusion that increased Focal adhesion signaling downregulates SEC23A levels which presumably modulates secretory traffic. I am not an expert on Focal adhesions but based on my understanding of the literature on that topic, downregulation of ROCK1, FERMT2 disrupts focal adhesions. (See: Theodosiou et. al., Elife, 2016 or Lock et. al., Plos One, 2012 for example). How do authors explain their results in siRNA knockdown of ROCK1 and FERMT2 which leads to an increased size of focal adhesions which seems contradictory to the published results? To clarify these results authors should test phosphorylation of FAK in their siRNA backgrounds which is another read out of focal adhesion signaling.

The experiments from cells grown on Fibronectin and Matrigel favor the argument which authors put forth, but authors may have to tweak the model a bit based on FAK phosphorylation and FAK signaling in context of above-mentioned knockdowns.

5)What happens to VSVG traffic or RUSH-Cadherin traffic when cells are plated on Matrigel and Fibronectin? Reduction in secretory traffic of these is an important experiment which is missing to close the loop and validate the model presented. Authors must test these experiments either with cells grown on matrix alone or in combination with siRNA to SEC23B. Authors should also monitor ERES and transport carriers in this background.

6)This is not such a major issue, but it would be good to see a comparison in SEC23A levels in siRNA knockdown condition in comparison to those when cells are grown on different substrates and in

ROCK1, FERMT2 knockdowns (blots of which authors already have in this manuscript).

****Minor comments:****

1)Scale bars are missing in EM images in Figure 2H.

2)Show molecular weight markers in Western blots in main figure 3E and supplementary figure S1E.

3. Significance:

Significance (Required)

I have looked at the manuscript from through the lens of a cell biologist as that is predominantly my area of expertise. In that respect I find the screen conducted by authors particularly interesting as they aim to connect how extracellular cues regulate the secretory pathway. A screen seems justified as there is no comprehensive understanding linking the two above-mentioned processes. Authors have done a functional interaction screen and analyzed a lot of images to identify candidates which either increase or decrease secretory traffic in combination with SEC23A and SEC23B. Such a functional screen has helped authors identify candidates which were otherwise missed in single siRNA knockdowns in their previous work from 2012. This definitely opens up interesting avenues to test the candidates identified in the screen in different physiological contexts and in disease as also the transcriptional program connecting Focal adhesion signaling with the regulation of components governing secretion. Such functional interaction screens could also be employed to identify crosstalk of different cellular processes with the regulation secretory pathway at ER as well as at the Golgi apparatus.

****Referees cross-commenting****

I agree with the comments from both the referees that the manuscript is very interesting, most experiments are well controlled, but the quantification of focal adhesion phenotype in knockdowns need to be done in an extensive manner and secretion phenotypes need to be measured upon plating cells on different matrix to validate the model presented.

Review #3

1. How much time do you estimate the authors will need to complete the suggested revisions:

Estimated time to Complete Revisions (Required)

(Decision Recommendation)

Between 1 and 3 months

2. Evidence, reproducibility and clarity:

Evidence, reproducibility and clarity (Required)

****Summary****

The authors use a synchronized cargo release assay following codepletion of either Sec23 paralog with cytoskeletal and associated proteins to identify potential functional interactions between COPII trafficking and the cytoskeleton. This screen yields a number of Sec23b

functionally interacting molecules that stall cargo trafficking to various degrees within the secretory pathway upon codepletion, and in the case of MACF1 reduce ERES number despite not physically interacting. Depletion of the majority of the identified Sec23b functional interactors alone surprisingly caused the downregulation of Sec23a at the mRNA and protein levels, and cargo trafficking could be partially or fully rescued by Sec23a overexpression depending on the codepleted cytoskeletal factor. RNA-seq enrichment analysis and imaging of a focal adhesion marker suggest that genes involved in cell adhesion were differentially regulated following depletion of the cytoskeletal functional interactors. Finally, the authors show that Sec23a expression levels are reduced when cells are cultured on dishes with high amounts of ECM to induce focal adhesions, and that inhibition of focal adhesion kinase can rescue Sec23a expression levels.

****Major comments****

The authors successfully implicate a group of cytoskeletal proteins and their actions at focal adhesions in negatively regulating Sec23a expression levels and COPII trafficking. This description of a shared, novel mode of COPII transcriptional regulation by cytoskeletal factors is convincingly shown to be at least a contributor to the delayed trafficking in the presence of focal adhesions. In general, the data are reproducible and use appropriate statistical analysis. However, a more robust description of the architecture of early secretory pathway would be beneficial, especially in the case of MACF1 codepletion which cannot be fully rescued by Sec23a-YFP overexpression. In contrast, trafficking during codepletion of FERMT2 is fully rescued by Sec23a-YFP despite both MACF1 and FERMT2 showing similar loss of Sec23a mRNA levels upon codepletion. This data suggests that while the trafficking delay in FERMT2 codepletion might be exclusively due to reduced Sec23a expression levels, there are likely additional causes for the trafficking delay observed in MACF1 codepletion.

While there is indeed a reduction in the number of ERESs following MACF1 codepletion, the authors report an even more dramatic reduction in 'transport intermediates / cell' as marked by COPI. However, as recent cryo-EM analysis of ERESs has definitively show, COPI exists stably at ERGIC membranes (1). Thus, an alternative possibility for the more dramatic reduction of COPI sites compared to Sec31a sites in Figures 2B-E is that ERGIC membranes are destabilized following MACF1 codepletion in a manner independent of Sec23a expression, and this destabilization compounds with reduced ERES number to ultimately delay trafficking. To more directly determine whether ERGIC membranes stability is regulated by MACF1, the authors should compare COPI and ERGIC-53 staining among MACF1 codepleted and FERMT2 codepleted cells with and without Sec23a-YFP overexpressed to levels that rescue cargo trafficking. If Sec23a-YFP restores the number of ERGIC punctae marked by these stains in FERMT2 but not MACF1 codepleted cells, it would suggest a role for MACF1 in forming or stabilizing ERGIC membranes which are known to associate with microtubules and WHAMM, an actin nucleator. Additionally, it would be useful to costain COPII with COPI or ERGIC-53 in control, MACF1 depleted, MACF1 codepleted, and MACF1 codepleted and Sec23a-YFP rescued cells to determine their colocalization. COPII and ERGIC membranes should be almost entirely coupled and juxtaposed in control cells and may be decoupled upon loss of MACF if plays a role in ERGIC membrane localization and stability. These proposed experiments are relevant because ERGIC membranes are sites of COPII cargo delivery and changes in ERGIC

stability or localization would suggest an additional mechanism for cytoskeletal regulation of COPII trafficking. These immunofluorescence studies should be straightforward and completed in a few weeks.

The choice to use VSVG and E-Cadherin for the synchronized release assays unfortunately convolutes interpreting the 'transport ratios' used by the authors to compare the effects of the various codepletions. Each protein progresses beyond the Golgi during secretion, and the authors choose to calculate the ratio of cargo intensity at the plasma membrane normalized to the total cellular cargo. This means that the synchronized release assays and calculated 'transport ratios' assay not only ER to Golgi trafficking, but also trafficking from the Golgi to the plasma membrane. In instances where Sec23a-YFP overexpression does not fully rescue the codepletion, it is possible that additional trafficking delays occur during Golgi to plasma membrane trafficking that cause the 'transport score' to decrease. Thus, the 'transport score' as the authors calculate it is needlessly nonspecific to COPII trafficking and should not be used to compare the codepletions for COPII functional interactors.

To mitigate unwanted contributions of post-COPII trafficking events from altering 'transport scores,' the authors should use a cargo for synchronized release assays that does not progress past the Golgi such as α -Mannosidase II and quantify a ratio of the perinuclear cargo signal to whole cell signal. Ideally, the screen would be repeated with a more appropriate cargo generating new 'transport scores' for the full list of cytoskeletal proteins. However, this may not be feasible, and as such 'transport scores' based on a Golgi resident protein should at least be produced for the 7 Sec23b functional interactors featured in this manuscript. These Golgi 'transport scores' would add much needed quantification of ER to Golgi transport delays that currently can only be inferred from the representative images in Figure 1E, which unfortunately show significant heterogeneity among cells from the same image. The authors should also explicitly state that any 'transport score' from a synchronous release assay using a cargo destined for the plasma membrane will take into account trafficking rate changes due not only to COPII, but also COPI from the ERGIC to the Golgi, and transport carriers departing from the TGN. These synchronized release assays would likely take between a few weeks to a few months depending on their ability to automate image analysis.

****Minor comments****

It would be useful for the authors to quantify the number of focal adhesions present from Vinculin stains from Figure 4C and 5C instead of just showing representative images. It would be interesting to determine if there is a meaningful relationship between focal adhesion number induced by the codepletions or tissue culture coating and Sec23a expression levels like in Figure 3D. Generally, the figures, text, and references were appropriate.

3. Significance:

Significance (Required)

In recent years, significant effort has been devoted to elucidating mechanisms by which COPII trafficking is modulated in response to cellular cues. These studies have revealed that changes in

nutrient availability, growth factors, ER stress, autophagy, and T-cell activation all cause changes in COPII trafficking via unique gene expression, splicing, or post-translational control (2-7). This work elucidates a novel mechanism of transcriptional control driven by focal adhesions. Additionally, it provides a number of potentially useful Sec23a and Sec23b functional interactors among cytoskeletal factors for further study. These unexplored factors may have unique mechanism of COPII regulation that could contribute to our understanding ER export modulation. Altogether, this and similar works are building an increasingly complex set of regulatory pathways that when integrated ultimately dictate COPII trafficking kinetics.

The reported findings are not only relevant to those who study COPII trafficking, but also other fields where secretion is studied in the context of the ECM. This work would suggest that secretion of factors involved in crosstalk between cells, including in tumors, is likely to be controlled by the interactions of cells with ECM.

Expertise keywords: cell biology, light microscopy, membrane trafficking

References

1. Weigel A V., Chang CL, Shtengel G, Xu CS, Hoffman DP, Freeman M, et al. ER-to-Golgi protein delivery through an interwoven, tubular network extending from ER. *Cell*. 2021 Apr;184(9):2412-2429.e16.
2. Farhan, H., Wendeler, M. W., Mitrovic, S., Fava, E., Silberberg, Y., Sharan, R., Zerial, M., & Hauri, H. P. (2010). MAPK signaling to the early secretory pathway revealed by kinase/phosphatase functional screening. *Journal of Cell Biology*, 189(6), 997-1011.
3. Zacharogianni, M., Kondylis, V., Tang, Y., Farhan, H., Xanthakis, D., Fuchs, F., Boutros, M., & Rabouille, C. (2011). ERK7 is a negative regulator of protein secretion in response to amino-acid starvation by modulating Sec16 membrane association. *EMBO Journal*, 30(18), 3684-3700.
4. Lillmann, K.D., V. Reiterer, F. Baschieri, J. Hoffmann, V. Millarte, M.A. Hauser, A. Mazza, N. Atias, D.F. Legler, R. Sharan, et al 2015. Regulation of Sec16 levels and dynamics links proliferation and secretion. *J. Cell Sci.* 128:670-682.
5. Liu, L., Cai, J., Wang, H., Liang, X., Zhou, Q., Ding, C., Zhu, Y., Fu, T., Guo, Q., Xu, Z., Xiao, L., Liu, J., Yin, Y., Fang, L., Xue, B., Wang, Y., Meng, Z. X., He, A., Li, J. L., ... Gan, Z. (2019). Coupling of COPII vesicle trafficking to nutrient availability by the IRE1 α -XBP1s axis. *Proceedings of the National Academy of Sciences of the United States of America*, 116(24), 11776-11785.
6. Jeong, Y.-T., Simoneschi, D., Keegan, S., Melville, D., Adler, N. S., Saraf, A., Florens, L., Washburn, M. P., Civasotto, C. N., Fenyö, D., Cuervo, A. M., Rossi, M., & Pagano, M. (2018). The ULK1-FBXW5-SEC23B nexus controls autophagy. *ELife*, 1-25.
7. Wilhelmi, I., Kanski, R., Neumann, A., Herdt, O., Hoff, F., Jacob, R., Preußner, M., & Heyd,

F. (2016). Sec16 alternative splicing dynamically controls COPII transport efficiency. *Nature Communications*, 7, 12347. <https://doi.org/10.1038/ncomms12347>

Manuscript number: RC-2021-01000

Corresponding author(s): Juan Jung, Rainer Pepperkok

[The “revision plan” should delineate the revisions that authors intend to carry out in response to the points raised by the referees. It also provides the authors with the opportunity to explain their view of the paper and of the referee reports.]

The document is important for the editors of affiliate journals when they make a first decision on the transferred manuscript. It will also be useful to readers of the reprint and help them to obtain a balanced view of the paper.

*If you wish to submit a full revision, please use our "Full Revision" template. **It is important to use the appropriate template to clearly inform the editors of your intentions.**]*

1. General Statements [optional]

We thank the reviewers for their critical comments and suggestions. We are glad that the reviewers appreciated the quality of the data and the novel findings connecting the secretory trafficking machinery with extracellular matrix-related signaling.

2. Description of the planned revisions

Reviewer #1 (Evidence, reproducibility and clarity (Required)):

The manuscript by Jung et al reports on an interesting finding that focal adhesion signaling regulates the expression of Sec23A and thereby regulates COPII-dependent trafficking. The data presented a mostly solid and the finding itself is highly novel, as it tackles an area of secretory trafficking that remains poorly understood, namely the connection between the ECM and secretion.

I will list below all comments that I have mixing both technical and conceptual topics:

Technical issues:

1-The authors should provide a better description of how the designed this siRNA library. What were the inclusion criteria for these 378 genes? I might have missed it, but I could not find this information easily.

Reply: The library has been designed in-house based on gene annotations and literature to include cytoskeleton structural proteins, motor proteins, and other associated and regulatory proteins. We will add this information in the Materials and Methods section.

2-Figure 2: I know this is challenging for EM images, but is there a way the authors could quantify these data? How many images were looked at? What was the average width of ER cisterne?

Reply: We will provide image quantifications and statistics

3-Figure 4: I think that the characterization of the FA phenotype is a bit underdeveloped. There is no quantification of these data. Is the size of FA changing? Is the number of FA per cell changing? Is the length of FAs changing? I think that more work is needed to increase the confidence in these data.

*I could also not easily see what type of cells these are. **A better description of this experiment is also required. Also, how many cells were analyzed.** I think it is important that this experiment is done with a sufficient number of cells to increase the confidence in the data.*

Reply: We agree with the reviewer that our observations regarding the focal adhesion (FA) phenotype will benefit from image quantification and we intend to include this in the revised manuscript. All FA experiments were performed on HeLa cells. We will update the materials and methods sections to better describe this experiment.

Conceptual issues:

1-The finding that focal adhesion signaling negatively affects ER-export is surprising, because cancer cells that grow on stiff substrates have more focal adhesions and are more invasive and migratory. Both migration and invasion are expected to depend on ER-export. Although the authors did not formally test Sec23A expression under different stiffnesses, I would expect that stiff substrates would lower Sec23A expression and thereby negatively affect ER-export. It would certainly increase the breadth of this work to include data like this and to also discuss this highly surprising finding. However, it is of course the decision of the authors and the editors to decide whether such an experiment would benefit the entire story.

Reply: In this work, we have shown that cells plated on ECM or matrigel have decreased SEC23A expression compared to control cells. We have also shown that inhibition of FA kinase leads to an increase in SEC23A expression (Figure 5). Whether this translates into a change in ER transport, is a fair point that we will address in the revision. Regarding stiffness, we have done a preliminary experiment that shows that cells plated on a soft synthetic substrate have

less SEC23A than cells plated on plastic. This goes in line with our ECM experiments because Matrigel and fibroblast-derived ECM are softer than plastic.

2-The authors postulate that this novel mechanism could be part of a feedback loop. If this were the case one would expect the acute effect of FA to increase ER-export (or secretion) and the negative feedback will then reduce secretion. However, the acute effect of FA is not addressed in this manuscript. In order to postulate a feedback loop, the authors would need to test the individual nodes of this loop.

Reply: The question appears to be whether an acute effect on FA would affect the expression of SEC23A and therefore ER transport. If by the acute effect the reviewer means a pharmacological manipulation, we have shown that upon treatment with the FAK inhibitor the expression of SEC23A increases (Fig 5A). Whether this increase in SEC23A expression translates into a corresponding increase in ER transport remains to be seen. This will be tested in our revised manuscript as mentioned above in reply to point # 1.

Our data encouraged us to propose a hypothetical feedback loop that would connect the deposition of ECM through the expression of SEC23A. We will have more data to support (or reject) this idea once we do the transport experiments as mentioned above. However, we think that a full characterization of this hypothetical loop by testing individual nodes is beyond the scope of this manuscript

Reviewer #1 (Significance (Required)):

I think that the basic finding of this manuscript is highly novel, by showing the impact of the ECM and focal adhesions on COPII-dependent trafficking. I think that this will not only appeal to people from the trafficking community, but also to people working on cell migration and on mechanobiology. The work in its current form does not require much extra efforts (max. 3 month). However, if the authors would decide to increase the breadth of data, they would require 3-6 months.

Reply: We thank reviewer #1 for the comments. We also believe that this story will appeal to a broader audience and would help to bridge the gap between membrane trafficking and mechanobiology communities.

Referees cross-commenting

I went through the comments of the two other reviewers and agree with their verdict. Some extra work on the characterization of the early secretory pathway would be good. Both reviewers provided a nice catalogue of possible experiments to choose from.

Reply: We have characterized the early secretory pathway in terms of ER exit sites, Beta-COP, and Golgi morphology (FIG. 2B-H and S1A-B). Together, these data strongly characterize the nature of ER-block. Moreover, the finding that our interactors affect the expression of SEC23A allows us to explain mechanistically why an ER transport block occurs. This is further strengthened by the rescue experiments (FIG. 3F). We believe that further characterization of the secretory pathway will not contribute substantially to the main message of this manuscript.

Reviewer #2 (Evidence, reproducibility and clarity (Required)):

The manuscript by Jung et al which based on a targeted siRNA screen, demonstrates regulation of SEC23A (component of the SEC23 complex of the COP coat) levels at transcriptional level downstream of focal adhesion signaling. By regulating siRNA mediated downregulation, the authors were able to identify proteins which either increased or decreased traffic of VSVG through the secretory pathway when combined with downregulation in the levels of with either SEC23A or SEC23B. Authors have focused on a group of SEC23B functional interactors, downregulation of which shows them increased size of focal adhesions which also downregulate SEC23A levels, thus providing an explanation for reduced secretory traffic. Authors further show that plating cells on fibronectin or Matrigel, which activate Focal adhesion kinase signaling also results in downregulation of SEC23A transcript levels. The screen is conducted in a well-controlled manner for most parts with a clear explanation of the analysis routines and the data presentation if of very good quality. Most important results have been validated by more than one experimental strategy which lends substantial confidence to the findings. The results also open further avenues for understanding the transcriptional regulation in different physiological and disease contexts.

There are certain issues, which the authors should address with regards to controls and some conflicting observations with published results with respect the phenotypes associated with downregulating proteins on focal adhesions size. Additionally, authors don't tie the ends by monitoring secretory traffic in cells grown on different matrices but include it in the model. Addressing/explaining these issues could improve this manuscript and the model may have to be tweaked a bit.

****Major comments:****

1)I wonder why the authors only used siRNA control in their screen when the effects are scored in context of double knockdown fashion in combination with mild knockdown of SEC23A and SEC23B to get functional interactors. Control siRNA in combination with SEC23A and SEC23B should have been two ideal negative controls in the screen. Nevertheless, in data presented Figure 1E and whole of Figure 2, using control siRNA in combination with SEC23B siRNA would have been ideal control to show that the combination does not induce any trafficking defects which could impact the findings of the study.

Hence, a few of the data presented from some of these figures should have sicontrol+SEC23B siRNA combination as a control.

Reply: There seems to be a misunderstanding. In the screen, the negative controls are only used as a reference as the scoring is based on a 5X5 matrix centered on the siRNA of interest. This is done to overcome possible plate effects and to normalize data across different biological replicas. As seen in figure 1B, the negative controls (Control siRNA or Control siRNA + SEC23A siRNA or Control siRNA + SEC23B siRNA) are very close to 0 (but not exactly 0) as they were not used in the normalization process. It is important to mention that all single knockdowns also contain our control siRNA to keep the same final siRNA concentration in single and double knockdowns. In Fig 1E we will include the images from Control + SEC23A siRNAs and Control + SEC23B siRNA as a reference. For Figure 2 all except 2A and 2H have the single knockdowns as controls.

2)What is the identity of post-ER structures which authors refer to in Figure 2A? Could the images represent VSVG concentrated at ER exit sites? Authors should stain with markers for ERES to see if the VSVG puncta colocalize with it.

Reply: We have done the experiment, and indeed these structures colocalize with an ER exit site marker (SEC31A). We intend to include this data into the revised manuscript. Our observations are in agreement with what is known in the literature about VSVG transport.

3)Based on RNA sequencing results, authors chose to follow up on SEC23A levels in background of siRNA knockdown of components (like MACF1, ROCK1, FERMT2 etc.) which regulate Focal adhesions in cells and show that there is a reduction in both transcript and protein levels of SEC23A. In images shown in Figure 2B and Figure 2C, levels of SEC31A and β -Cop1 are reduced. Authors should test using qPCR and western blots whether there is a downregulation SEC31A, β -Cop1 and SEC23B in siRNA knockdowns of MACF1, ROCK1, FERMT2 etc. It would provide new insights if there were a co-regulation of secretory machinery to modulate the secretory traffic in response to Focal Adhesion based signaling.

Reply: Our transcriptomics data (FIG 3C and Table 5) shows that SEC31A and COPB1 mRNAs are not altered upon any of the knockdowns. For SEC23B, we observed only a slight decrease in ROCK1 knockdown. This data suggests that a co-regulation of the secretory machinery might not be present. Instead, the curation of secretory pathway genes in our transcriptome data shows that SEC23A is the only commonly differentially expressed gene.

4)Most major concern in this manuscript surrounds around results presented in Figure 4C. Authors show that in response to all the knockdowns, they see more focal adhesions as monitored by Vinculin staining and this along with the experiments with cells plated on Matrigel and Fibronectin arrive at the conclusion that increased Focal adhesion signaling downregulates SEC23A levels which presumably modulates secretory traffic. I am not an expert on Focal adhesions but based on my understanding of the literature on that topic,

downregulation of ROCK1, FEMRT2 disrupts focal adhesions. (See: Theodosiou et. al., Elife, 2016 or Lock et. al., Plos One, 2012 for example). How do authors explain their results in siRNA knockdown of ROCK1 and FEMRT2 which leads to an increased size of focal adhesions which seems contradictory to the published results? To clarify these results authors should test phosphorylation of FAK in their siRNA backgrounds which is another read out of focal adhesion signaling.

The experiments from cells grown on Fibronectin and Matrigel favor the argument which authors put forth, but authors may have to tweak the model a bit based on FAK phosphorylation and FAK signaling in context of above-mentioned knockdowns.

Reply: Based on the images for vinculin staining, in our current manuscript we propose that changes in FAs occur upon knocking down our interactors. In our revised manuscript we will provide a more robust quantitative assessment of those changes (change in number, size, or intensity) as mentioned in our reply to Reviewer #1.

As for the discrepancies in the relation of FA phenotype upon depletion of ROCK1 and FERMT2, we want to point out that this effect depends on the cell type used. For instance, the papers listed by the reviewer here use fibroblasts and keratinocytes respectively while we have used Hela Kyoto cells which are epithelial in nature. Another example is that while in fibroblasts depletion of FERMT2 leads to a rounded morphology and almost an absence of FAs (*Theodosiou et. al., Elife, 2016*), in podocytes (Qu et al JCS, 2011), it leads to fewer FAs but an increase in their size. Nonetheless, this is a very keen observation from the reviewer and we will address this point in our revised manuscript discussion.

5)What happens to VSVG traffic or RUSH-Cadherin traffic when cells are plated on Matrigel and Fibronectin? Reduction in secretory traffic of these is an important experiment which is missing to close the loop and validate the model presented. Authors must test these experiments either with cells grown on matrix alone or in combination with siRNA to SEC23B. Authors should also monitor ERES and transport carriers in this background.

Reply: We agree with the reviewer and intend to perform these experiments.

6)This is not such a major issue, but it would be good to see a comparison in SEC23A levels in siRNA knockdown condition in comparison to those when cells are grown on different substrates and in ROCK1, FEMRT2 knockdowns (blots of which authors already have in this manuscript).

Reply: We will assess the level of SEC23A at the protein level for cells plated on matrigel or Fibroblast-derived ECM.

****Minor comments:****

1)Scale bars are missing in EM images in Figure 2H.

Reply: We will add the scales in our EM images

2)Show molecular weight markers in Western blots in main figure 3E and supplementary figure S1E.

Reply: We will add molecular weight markers in our Western-Blots

Reviewer #2 (Significance (Required)):

I have looked at the manuscript from through the lens of a cell biologist as that is predominantly my area of expertise. In that respect I find the screen conducted by authors particularly interesting as they aim to connect how extracellular cues regulate the secretory pathway. A screen seems justified as there is no comprehensive understanding linking the two above-mentioned processes. Authors have done a functional interaction screen and analyzed a lot of images to identify candidates which either increase or decrease secretory traffic in combination with SEC23A and SEC23B. Such a functional screen has helped authors identify candidates which were otherwise missed in single siRNA knockdowns in their previous work from 2012. This definitely opens up interesting avenues to test the candidates identified in the screen in different physiological contexts and in disease as also the transcriptional program connecting Focal adhesion signaling with the regulation of components governing secretion. Such functional interaction screens could also be employed to identify crosstalk of different cellular processes with the regulation secretory pathway at ER as well as at the Golgi apparatus.

Reply: We thank reviewer #2 for the comments. As we mentioned in our reply to reviewer #1, we strongly believe that these results will encourage further research at the crossroads of membrane trafficking and mechanobiology.

Referees cross-commenting

I agree with the comments from both the referees that the manuscript is very interesting, most experiments are well controlled, but the quantification of focal adhesion phenotype in knockdowns need to be done in an extensive manner and secretion phenotypes need to be measured upon plating cells on different matrix to validate the model presented.

Revision Plan

Reply: These two experiments will be included in our revision

Reviewer #3 (Evidence, reproducibility and clarity (Required)):

****Summary****

The authors use a synchronized cargo release assay following codepletion of either Sec23 paralog with cytoskeletal and associated proteins to identify potential functional interactions between COPII trafficking and the cytoskeleton. This screen yields a number of Sec23b functionally interacting molecules that stall cargo trafficking to various degrees within the secretory pathway upon codepletion, and in the case of MACF1 reduce ERES number despite not physically interacting. Depletion of the majority of the identified Sec23b functional interactors alone surprisingly caused the downregulation of Sec23a at the mRNA and protein levels, and cargo trafficking could be partially or fully rescued by Sec23a overexpression depending on the codepleted cytoskeletal factor. RNA-seq enrichment analysis and imaging of a focal adhesion marker suggest that genes involved in cell adhesion were differentially regulated following depletion of the cytoskeletal functional interactors. Finally, the authors show that Sec23a expression levels are reduced when cells are cultured on dishes with high amounts of ECM to induce focal adhesions, and that inhibition of focal adhesion kinase can rescue Sec23a expression levels.

****Major comments****

#1 The authors successfully implicate a group of cytoskeletal proteins and their actions at focal adhesions in negatively regulating Sec23a expression levels and COPII trafficking. This description of a shared, novel mode of COPII transcriptional regulation by cytoskeletal factors is convincingly shown to be at least a contributor to the delayed trafficking in the presence of focal adhesions. In general, the data are reproducible and use appropriate statistical analysis. However, a more robust description of the architecture of early secretory pathway would be beneficial, especially in the case of MACF1 codepletion which cannot be fully rescued by Sec23a-YFP overexpression. In contrast, trafficking during codepletion of FERMT2 is fully rescued by Sec23a-YFP despite both MACF1 and FERMT2 showing similar loss of Sec23a mRNA levels upon codepletion. This data suggests that while the trafficking delay in FERMT2 codepletion might be exclusively due to reduced Sec23a expression levels, there are likely additional causes for the trafficking delay observed in MACF1 codepletion.

Reply: We thank the reviewer for the appreciation of our results and the importance they might bear for the field. The reviewer has very neatly highlighted that each of our interactor hits might have roles in the secretory pathway beyond the ER or independent of the expression levels of SEC23A. This phenomenon could also explain the differential rescue of the arrival of VSVG at the plasma membrane upon SEC23A overexpression in FERMT2 and MACF1 knockdowns (FIG

3F). For instance, MACF1 has been involved in Golgi to Plasma Membrane transport as well (Kakinuma et al. Exp. Cell Res. 2004, Burgo et al. Dev. Cell 2012). So a possibility is that SEC23A overexpression rescues only ER to Golgi transport but the lack of rescue in the compartment between Golgi and plasma membrane independent of SEC23A expression levels would result in reduced rescue in the case of MACF1 compared to FERMT2. To support this, in our revised manuscript, we will provide example images from the experiment.

Nonetheless, we agree that these are very important observations from Reviewer #3 and warrant a detailed discussion in the light of other interactors as well, which we intend to highlight in our revised manuscript.

#2 While there is indeed a reduction in the number of ERESs following MACF1 codepletion, the authors report an even more dramatic reduction in 'transport intermediates / cell' as marked by COPI. However, as recent cryo-EM analysis of ERESs has definitively show, COPI exists stably at ERGIC membranes (1). Thus, an alternative possibility for the more dramatic reduction of COPI sites compared to Sec31a sites in Figures 2B-E is that ERGIC membranes are destabilized following MACF1 codepletion in a manner independent of Sec23a expression, and this destabilization compounds with reduced ERES number to ultimately delay trafficking. To more directly determine whether ERGIC membranes stability is regulated by MACF1, the authors should compare COPI and ERGIC-53 staining among MACF1 codepleted and FERMT2 codepleted cells with and without Sec23a-YFP overexpressed to levels that rescue cargo trafficking. If Sec23a-YFP restores the number of ERGIC punctae marked by these stains in FERMT2 but not MACF1 codepleted cells, it would suggest a role for MACF1 in forming or stabilizing ERGIC membranes which are known to associate with microtubules and WHAMM, an actin nucleator. Additionally, it would be useful to costain COPII with COPI or ERGIC-53 in control, MACF1 depleted, MACF1 codepleted, and MACF1 codepleted and Sec23a-YFP rescued cells to determine their colocalization. COPII and ERGIC membranes should be almost entirely coupled and juxtaposed in control cells and may be decoupled upon loss of MACF if plays a role in ERGIC membrane localization and stability. These proposed experiments are relevant because ERGIC membranes are sites of COPII cargo delivery and changes in ERGIC stability or localization would suggest an additional mechanism for cytoskeletal regulation of COPII trafficking. These immunofluorescence studies should be straightforward and completed in a few weeks.

Reply: Although a possible additional role of MACF1 in the organisation of early secretory pathway, stability of ERES, etc., independent of the expression of SEC23A is interesting on its own, we believe that an extensive characterization of these possible roles/ pathways as proposed by the reviewer is beyond the scope this manuscript.

#3 The choice to use VSVG and E-Cadherin for the synchronized release assays unfortunately convolutes interpreting the 'transport ratios' used by the authors to compare the effects of the various codepletions. Each protein progresses beyond the Golgi during secretion, and the authors choose to calculate the ratio of cargo intensity at the plasma membrane normalized to the total cellular cargo. This means that the synchronized release assays and calculated 'transport ratios' assay not only ER to Golgi trafficking, but also

trafficking from the Golgi to the plasma membrane. In instances where Sec23a-YFP overexpression does not fully rescue the codepletion, it is possible that additional trafficking delays occur during Golgi to plasma membrane trafficking that cause the 'transport score' to decrease. Thus, the 'transport score' as the authors calculate it is needlessly nonspecific to COPII trafficking and should not be used to compare the codepletions for COPII functional interactors.

Reply: We agree that the “transport score” used here and in our previous genome-wide screen (Simpson et. al Nat. Cell Biol. 2012) does not allow us to distinguish between the individual transport substeps in the transport of VSVG from the ER to the plasma membrane. However, as we see in Fig 1E, the proteins that we have decided to follow in more detail in this study do have a clear ER transport block phenotype (except for CRKL). So for 6 out of 7 of these proteins, the images clearly show that the decrease in the “transport score” is due to a decreased ER to Golgi transport.

#4 To mitigate unwanted contributions of post-COPII trafficking events from altering 'transport scores,' the authors should use a cargo for synchronized release assays that does not progress past the Golgi such as α -Mannosidase II and quantify a ratio of the perinuclear cargo signal to whole cell signal. Ideally, the screen would be repeated with a more appropriate cargo generating new 'transport scores' for the full list of cytoskeletal proteins. However, this may not be feasible, and as such 'transport scores' based on a Golgi resident protein should at least be produced for the 7 Sec23b functional interactors featured in this manuscript. These Golgi 'transport scores' would add much needed quantification of ER to Golgi transport delays that currently can only be inferred from the representative images in Figure 1E, which unfortunately show significant heterogeneity among cells from the same image. The authors should also explicitly state that any 'transport score' from a synchronous release assay using a cargo destined for the plasma membrane will take into account trafficking rate changes due not only to COPII, but also COPI from the ERGIC to the Golgi, and transport carriers departing from the TGN. These synchronized release assays would likely take between a few weeks to a few months depending on their ability to automate image analysis.

Reply: We consider that having a “Golgi transport score” won't add any new information as the proteins that we have chosen to follow are the ones that show a strong ER-block phenotype. However, we agree that such a “Golgi score” would indeed be useful if one would like to study other interactors, for instance, the ones that induce transport acceleration.

Also, we don't expect all cells to behave similarly as the level of knockdown might be slightly different or because of the cell to cell variability. Even in control conditions (no knockdown), this heterogeneity is evident. As suggested by the reviewer, in our revised manuscript we will explicitly state that a change in the transport scores could mean a change in any sub-step of the transport from the ER to the PM in our assay.

****Minor comments****

It would be useful for the authors to quantify the number of focal adhesions present from Vinculin stains from Figure 4C and 5C instead of just showing representative images. It would be interesting to determine if there is a meaningful relationship between focal adhesion number induced by the codepletions or tissue culture coating and Sec23a expression levels like in Figure 3D. Generally, the figures, text, and references were appropriate.

Reply: As also pointed out by the other reviewers we will quantify the FA changes

Reviewer #3 (Significance (Required)):

In recent years, significant effort has been devoted to elucidating mechanisms by which COPII trafficking is modulated in response to cellular cues. These studies have revealed that changes in nutrient availability, growth factors, ER stress, autophagy, and T-cell activation all cause changes in COPII trafficking via unique gene expression, splicing, or post-translational control (2-7). This work elucidates a novel mechanism of transcriptional control driven by focal adhesions. Additionally, it provides a number of potentially useful Sec23a and Sec23b functional interactors among cytoskeletal factors for further study. These unexplored factors may have unique mechanism of COPII regulation that could contribute to our understanding ER export modulation. Altogether, this and similar works are building an increasingly complex set of regulatory pathways that when integrated ultimately dictate COPII trafficking kinetics.

The reported findings are not only relevant to those who study COPII trafficking, but also other fields where secretion is studied in the context of the ECM. This work would suggest that secretion of factors involved in crosstalk between cells, including in tumors, is likely to be controlled by the interactions of cells with ECM.

Reply: We thank reviewer #3 for the comments and insightful discussion about the limitations of our assay that we will highlight in the revised manuscript and in general for the insight into the early secretory pathway regulation. Furthermore their explicit summary of how our study could bridge COPII trafficking, ECM signaling and the relevance to various pathophysiologies is highly appreciated.

Expertise keywords: cell biology, light microscopy, membrane trafficking

References

1. Weigel A V., Chang CL, Shtengel G, Xu CS, Hoffman DP, Freeman M, et al. ER-to-Golgi protein delivery through an interwoven, tubular network extending from ER. Cell. 2021 Apr;184(9):2412-2429.e16.

2. Farhan, H., Wendeler, M. W., Mitrovic, S., Fava, E., Silberberg, Y., Sharan, R., Zerial, M., & Hauri, H. P. (2010). MAPK signaling to the early secretory pathway revealed by kinase/phosphatase functional screening. *Journal of Cell Biology*, 189(6), 997-1011.

3. Zacharogianni, M., Kondylis, V., Tang, Y., Farhan, H., Xanthakis, D., Fuchs, F., Boutros, M., & Rabouille, C. (2011). ERK7 is a negative regulator of protein secretion in response to amino-acid starvation by modulating Sec16 membrane association. *EMBO Journal*, 30(18), 3684-3700.

4. Lillmann, K.D., V. Reiterer, F. Baschieri, J. Hoffmann, V. Millarte, M.A. Hauser, A. Mazza, N. Atias, D.F. Legler, R. Sharan, et al 2015. Regulation of Sec16 levels and dynamics links proliferation and secretion. *J. Cell Sci.* 128:670-682.

5. Liu, L., Cai, J., Wang, H., Liang, X., Zhou, Q., Ding, C., Zhu, Y., Fu, T., Guo, Q., Xu, Z., Xiao, L., Liu, J., Yin, Y., Fang, L., Xue, B., Wang, Y., Meng, Z. X., He, A., Li, J. L., ... Gan, Z. (2019). Coupling of COPII vesicle trafficking to nutrient availability by the IRE1 α -XBP1s axis. *Proceedings of the National Academy of Sciences of the United States of America*, 116(24), 11776-11785.

6. Jeong, Y.-T., Simoneschi, D., Keegan, S., Melville, D., Adler, N. S., Saraf, A., Florens, L., Washburn, M. P., Cavasotto, C. N., Fenyö, D., Cuervo, A. M., Rossi, M., & Pagano, M. (2018). The ULK1-FBXW5-SEC23B nexus controls autophagy. *ELife*, 1-25.

7. Wilhelmi, I., Kanski, R., Neumann, A., Herdt, O., Hoff, F., Jacob, R., Preußner, M., & Heyd, F. (2016). Sec16 alternative splicing dynamically controls COPII transport efficiency. *Nature Communications*, 7, 12347. <https://doi.org/10.1038/ncomms12347>

3. Description of the revisions that have already been incorporated in the transferred manuscript

4. Description of analyses that authors prefer not to carry out

Reviewer #3 suggested to robustly characterise the early secretory pathway, in response to the depletion of our interactors, for instance, the role of MACF1 in the organization and the stability of ERES. This view is also supported by reviewer #1. However, in our revised manuscript we would like to focus more on the novel aspect of our study (as highlighted by all the reviewers), namely how ECM signaling and changes in FAs affect SEC23A and possibly ER transport. For this, we would like to present a more quantitative outlook of the FA phenotype and concentrate on the transport experiments. The reason for not dwelling into a more extensive characterization of the early secretory pathway is that these experiments are very interesting on their own, and

Revision Plan

merit a separate study that would deconvolve in detail the individual trafficking steps, and their relation to SEC23A expression, ERES stability, and ECM signaling.

Reviewer #2 suggested that to better characterize the FA phenotype and solve the apparent discrepancies between our data and the literature, we could test FAK phosphorylation. As we mentioned in our reply to this point, we think that most of the discrepancies arise from the different cell types used. Nevertheless, we agree that a quantitative approach is needed for a better characterisation of FA phenotype, therefore we intend to perform quantification of the vinculin stainings.

October 18, 2021

Re: JCB manuscript #202110081T

Dr. Juan Jung
European Molecular Biology Laboratory
Cell Biology & Biophysics
Meyerhofstrasse 1
Heidelberg 69117
Germany

Dear Dr. Jung,

Thank you for submitting your manuscript entitled "A high throughput SEC23 functional interaction screen reveals a role for focal adhesion and extracellular matrix signalling in the regulation of COPII subunit SEC23A" from Review Commons. We find your study potentially appropriate as a JCB Report and overall agree with your revision plan. In particular we find the reviewer comments that a more careful quantification of focal adhesions as well as an analysis of different substrates is essential, therefore please ensure that a revised submission completely addresses these points. While we think a more detailed investigation into the changes in the early secretory pathway as suggested by Reviewer #3 would be interesting and would of course welcome such data, we accept this is beyond the scope. While this issue does not need to be experimentally addressed, you should however discuss the reviewer's point that cross-talk or interplay between ERES/COPII function and ERGIC/COPI function is likely linked and therefore of importance in interpreting some of your observations.

GENERAL GUIDELINES:

Text limits: Character count for a Transfer is < 20,000, not including spaces. Count includes title page, abstract, introduction, results, discussion, acknowledgments, and figure legends. Count does not include materials and methods, references, tables, or supplemental legends.

Figures: Transfers may have up to 5 main text figures. To avoid delays in production, figures must be prepared according to the policies outlined in our Instructions to Authors, under Data Presentation, <https://jcb.rupress.org/site/misc/ifora.xhtml>. All figures in accepted manuscripts will be screened prior to publication.

Supplemental information: There are strict limits on the allowable amount of supplemental data. Transfers may have up to 3 supplemental figures. Up to 10 supplemental videos or flash animations are allowed. A summary of all supplemental material should appear at the end of the Materials and methods section.

Please note that JCB now requires authors to submit Source Data used to generate figures containing gels and Western blots with all revised manuscripts. This Source Data consists of fully uncropped and unprocessed images for each gel/blot displayed in the main and supplemental figures. Since your paper includes cropped gel and/or blot images, please be sure to provide one Source Data file for each figure that contains gels and/or blots along with your revised manuscript files. File names for Source Data figures should be alphanumeric without any spaces or special characters (i.e., SourceDataF#, where F# refers to the associated main figure number or SourceDataFS# for those associated with Supplementary figures). The lanes of the gels/blots should be labeled as they are in the associated figure, the place where cropping was applied should be marked (with a box), and molecular weight/size standards should be labeled wherever possible. Source Data files will be made available to reviewers during evaluation of revised manuscripts and, if your paper is eventually published in JCB, the files will be directly linked to specific figures in the published article.

As you may know, the typical timeframe for revisions is three to four months. However, we at JCB realize that the implementation of social distancing measures that limit spread of COVID-19 also pose challenges to scientific researchers. Therefore, JCB has waived the revision time limit. Please note that papers are generally considered through only one revision

cycle, so any revised manuscript will likely be either accepted or rejected.

Thank you for this interesting contribution to Journal of Cell Biology. You can contact us at the journal office with any questions, cellbio@rockefeller.edu or call (212) 327-8588.

Sincerely,

Elizabeth Miller, PhD
Monitoring Editor

Andrea L. Marat, PhD
Senior Scientific Editor

Journal of Cell Biology

Revision Plan

Manuscript number: RC-2021-01000

Corresponding author(s): Juan Jung, Rainer Pepperkok

[The “revision plan” should delineate the revisions that authors intend to carry out in response to the points raised by the referees. It also provides the authors with the opportunity to explain their view of the paper and of the referee reports.]

The document is important for the editors of affiliate journals when they make a first decision on the transferred manuscript. It will also be useful to readers of the reprint and help them to obtain a balanced view of the paper.

*If you wish to submit a full revision, please use our "Full Revision" template. **It is important to use the appropriate template to clearly inform the editors of your intentions.**]*

1. General Statements [optional]

We thank the reviewers for their critical comments and suggestions. We are glad that the reviewers appreciated the quality of the data and the novel findings connecting the secretory trafficking machinery with extracellular matrix-related signaling.

2. Description of the planned revisions

Reviewer #1 (Evidence, reproducibility and clarity (Required)):

The manuscript by Jung et al reports on an interesting finding that focal adhesion signaling regulates the expression of Sec23A and thereby regulates COPII-dependent trafficking. The data presented a mostly solid and the finding itself is highly novel, as it tackles an area of secretory trafficking that remains poorly understood, namely the connection between the ECM and secretion.

I will list below all comments that I have mixing both technical and conceptual topics:

****Technical issues:****

1-The authors should provide a better description of how the designed this siRNA library. What were the inclusion criteria for these 378 genes? I might have missed it, but I could not find this information easily.

Reply: The library has been designed in-house based on gene annotations and literature to include cytoskeleton structural proteins, motor proteins, and other associated and regulatory proteins. We will add this information in the Materials and Methods section.

Revision Plan

Update: We have included a more detailed description of the library. See Material and Methods (section siRNA-based functional interaction screen)

2-Figure 2: I know this is challenging for EM images, but is there a way the authors could quantify these data? How many images were looked at? What was the average width of ER cisterne?

Reply: We will provide image quantifications and statistics

Update: We have quantified the EM images. The quantifications are shown in Figure S1E

3-Figure 4: I think that the characterization of the FA phenotype is a bit underdeveloped. There is no quantification of these data. Is the size of FA changing? Is the number of FA per cell changing? Is the length of FAs changing? I think that more work is needed to increase the confidence in these data.

*I could also not easily see what type of cells these are. **A better description of this experiment is also required. Also, how many cells were analyzed.** I think it is important that this experiment is done with a sufficient number of cells to increase the confidence in the data.*

Reply: We agree with the reviewer that our observations regarding the focal adhesion (FA) phenotype will benefit from image quantification and we intend to include this in the revised manuscript. All FA experiments were performed on HeLa cells. We will update the materials and methods sections to better describe this experiment.

Update: We have quantified the FA phenotype (see Figure 4C). Indeed, our quantification confirms that in all the tested knockdowns (hits), there are morphological alterations in FAs. Adequate explanation is presented in the results section. We have added a detailed description of image analysis (number of cells quantified, cell type) in the M&M section.

****Conceptual issues:****

1-The finding that focal adhesion signaling negatively affects ER-export is surprising, because cancer cells that grow on stiff substrates have more focal adhesions and are more invasive and migratory. Both migration and invasion are expected to depend on ER-export. Although the authors did not formally test Sec23A expression under different stiffnesses, I would expect that stiff substrates would lower Sec23A expression and thereby negatively affect ER-export. It would certainly increase the breadth of this work to include data like

Revision Plan

this and to also discuss this highly surprising finding. However, it is of course the decision of the authors and the editors to decide whether such an experiment would benefit the entire story.

Reply: In this work, we have shown that cells plated on ECM or Matrigel have decreased SEC23A expression compared to control cells. We have also shown that inhibition of FA kinase leads to an increase in SEC23A expression (Figure 5). Whether this translates into a change in ER transport, is a fair point that we will address in the revision. Regarding stiffness, we have done a preliminary experiment that shows that cells plated on a soft synthetic substrate have less SEC23A than cells plated on plastic. This goes in line with our ECM experiments because Matrigel and fibroblast-derived ECM are softer than plastic

Update: We have included VSVG transport experiment in cells plated on plastic and Matrigel. The results show that there is indeed a decrease in the transport of VSVG when cells are plated on Matrigel compared to plastic substrate. This phenotype was further rescued after treatment with Focal Adhesion kinase inhibitor, confirming the role of FA and substrate nature on cargo transport. The data is shown in Figure 5E.

2-The authors postulate that this novel mechanism could be part of a feedback loop. If this were the case one would expect the acute effect of FA to increase ER-export (or secretion) and the negative feedback will then reduce secretion. However, the acute effect of FA is not addressed in this manuscript. In order to postulate a feedback loop, the authors would need to test the individual nodes of this loop.

Reply: The question appears to be whether an acute effect on FA would affect the expression of SEC23A and therefore ER transport. If by the acute effect the reviewer means a pharmacological manipulation, we have shown that upon treatment with the FAK inhibitor the expression of SEC23A increases (Fig 5A). Whether this increase in SEC23A expression translates into a corresponding increase in ER transport remains to be seen. This will be tested in our revised manuscript as mentioned above in reply to point # 1.

Our data encouraged us to propose a hypothetical feedback loop that would connect the deposition of ECM through the expression of SEC23A. We will have more data to support (or reject) this idea once we do the transport experiments as mentioned above. However, we think that a full characterization of this hypothetical loop by testing individual nodes is beyond the scope of this manuscript

Reviewer #1 (Significance (Required)):

I think that the basic finding of this manuscript is highly novel, by showing the impact of the ECM and focal adhesions on COPII-dependent trafficking. I think that this will not only appeal to people from the trafficking community, but also to people working on cell migration and on mechanobiology. The work in its

Revision Plan

current form does not require much extra efforts (max. 3 month). However, if the authors would decide to increase the breadth of data, they would require 3-6 months.

Reply: We thank reviewer #1 for the comments. We also believe that this story will appeal to a broader audience and would help to bridge the gap between membrane trafficking and mechanobiology communities.

Referees cross-commenting

I went through the comments of the two other reviewers and agree with their verdict. Some extra work on the characterization of the early secretory pathway would be good. Both reviewers provided a nice catalogue of possible experiments to choose from.

Reply: We have characterized the early secretory pathway in terms of ER exit sites, Beta-COP, and Golgi morphology (FIG. 2B-H and S1A-B). Together, these data strongly characterize the nature of ER-block. Moreover, the finding that our interactors affect the expression of SEC23A allows us to explain mechanistically why an ER transport block occurs. This is further strengthened by the rescue experiments (FIG. 3F). We believe that further characterization of the secretory pathway will not contribute substantially to the main message of this manuscript.

Reviewer #2 (Evidence, reproducibility and clarity (Required)):

The manuscript by Jung et al which based on a targeted siRNA screen, demonstrates regulation of SEC23A (component of the SEC23 complex of the COP coat) levels at transcriptional level downstream of focal adhesion signaling. By regulating siRNA mediated downregulation, the authors were able to identify proteins which either increased or decreased traffic of VSVG through the secretory pathway when combined with downregulation in the levels of with either SEC23A or SEC23B. Authors have focused on a group of SEC23B functional interactors, downregulation of which shows them increased size of focal adhesions which also downregulate SEC23A levels, thus providing an explanation for reduced secretory traffic. Authors further show that plating cells on fibronectin or Matrigel, which activate Focal adhesion kinase signaling also results in downregulation of SEC23A transcript levels. The screen is conducted in a well-controlled manner for most parts with a clear explanation of the analysis routines and the data presentation if of very good quality. Most important results have been validated by more than one experimental strategy which lends substantial confidence to the findings. The results also open further avenues for understanding the transcriptional regulation in different physiological and disease contexts.

There are certain issues, which the authors should address with regards to controls and some conflicting observations with published results with respect the phenotypes associated with downregulating proteins on focal adhesions

Revision Plan

size. Additionally, authors don't tie the ends by monitoring secretory traffic in cells grown on different matrices but include it in the model. Addressing/explaining these issues could improve this manuscript and the model may have to be tweaked a bit.

****Major comments:****

1)I wonder why the authors only used siRNA control in their screen when the effects are scored in context of double knockdown fashion in combination with mild knockdown of SEC23A and SEC23B to get functional interactors. Control siRNA in combination with SEC23A and SEC23B should have been two ideal negative controls in the screen. Nevertheless, in data presented Figure 1E and whole of Figure 2, using control siRNA in combination with SEC23B siRNA would have been ideal control to show that the combination does not induce any trafficking defects which could impact the findings of the study. Hence, a few of the data presented from some of these figures should have sicontrol+SEC23B siRNA combination as a control.

Reply: There seems to be a misunderstanding. In the screen, the negative controls are only used as a reference as the scoring is based on a 5X5 matrix centered on the siRNA of interest. This is done to overcome possible plate effects and to normalize data across different biological replicas. As seen in figure 1B, the negative controls (Control siRNA or Control siRNA + SEC23A siRNA or Control siRNA + SEC23B siRNA are very close to 0 (but not exactly 0) as they were not used in the normalization process. It is important to mention that all single knockdowns also contain our control siRNA to keep the same final siRNA concentration in single and double knockdowns. In Fig 1E we will include the images from Control + SEC23A siRNAs and Control + SEC23B siRNA as a reference. For Figure 2 all except 2A and 2H have the single knockdowns as controls.

2)What is the identity of post-ER structures which authors refer to in Figure 2A? Could the images represent VSVG concentrated at ER exit sites? Authors should stain with markers for ERES to see if the VSVG puncta colocalize with it.

Reply: We have done the experiment, and indeed these structures colocalize with an ER exit site marker (SEC31A). We intend to include this data into the revised manuscript. Our observations are in agreement with what is known in the literature about VSVG transport.

Update: Since the further characterisation of the secretory pathway is not a major point that needs further characterisation and we have discussed this at various appropriate places throughout the manuscript. Therefore, we have not included this data anymore.

3)Based on RNA sequencing results, authors chose to follow up on SEC23A levels in background of siRNA knockdown of components (like MACF1, ROCK1, FERMT2 etc.) which regulate Focal adhesions in cells and show that

there is a reduction in both transcript and protein levels of SEC23A. In images shown in Figure 2B and Figure 2C, levels of SEC31A and β -Cop1 are reduced. Authors should test using qPCR and western blots whether there is a downregulation of SEC31A, β -Cop1 and SEC23B in siRNA knockdowns of MACF1, ROCK1, FERMT2 etc. It would provide new insights if there were a co-regulation of secretory machinery to modulate the secretory traffic in response to Focal Adhesion based signaling.

Reply: Our transcriptomics data (FIG 3C and Table 5) shows that SEC31A and COPB1 mRNAs are not altered upon any of the knockdowns. For SEC23B, we observed only a slight decrease in ROCK1 knockdown. This data suggests that a co-regulation of the secretory machinery might not be present. Instead, the curation of secretory pathway genes in our transcriptome data shows that SEC23A is the only commonly differentially expressed gene.

4) Most major concern in this manuscript surrounds around results presented in Figure 4C. Authors show that in response to all the knockdowns, they see more focal adhesions as monitored by Vinculin staining and this along with the experiments with cells plated on Matrigel and Fibronectin arrive at the conclusion that increased Focal adhesion signaling downregulates SEC23A levels which presumably modulates secretory traffic. I am not an expert on Focal adhesions but based on my understanding of the literature on that topic, downregulation of ROCK1, FERMT2 disrupts focal adhesions. (See: Theodosiou et. al., Elife, 2016 or Lock et. al., Plos One, 2012 for example). How do authors explain their results in siRNA knockdown of ROCK1 and FERMT2 which leads to an increased size of focal adhesions which seems contradictory to the published results? To clarify these results authors should test phosphorylation of FAK in their siRNA backgrounds which is another read out of focal adhesion signaling.

The experiments from cells grown on Fibronectin and Matrigel favor the argument which authors put forth, but authors may have to tweak the model a bit based on FAK phosphorylation and FAK signaling in context of above-mentioned knockdowns.

Reply: Based on the images for vinculin staining, in our current manuscript we propose that changes in FAs occur upon knocking down our interactors. In our revised manuscript we will provide a more robust quantitative assessment of those changes (change in number, size, or intensity) as mentioned in our reply to Reviewer #1.

As for the discrepancies in the relation of FA phenotype upon depletion of ROCK1 and FERMT2, we want to point out that this effect depends on the cell type used. For instance, the papers listed by the reviewer here use fibroblasts and keratinocytes respectively while we have used HeLa Kyoto cells which are epithelial in nature. Another example is that while in fibroblasts depletion of FERMT2 leads to a rounded morphology and almost an absence of FAs (*Theodosiou et. al., Elife, 2016*), in podocytes (Qu et al JCS, 2011), it leads to fewer FAs but an increase in their size. Nonetheless, this is a very keen observation from the reviewer and we will address this point in our revised manuscript discussion.

Revision Plan

Update: Based on our quantifications of the FA phenotype upon depletion of the interactors we concluded that there are changes in FA morphology for all the knockdowns tested. Some of these changes have not been previously described and some might seem contradictory with the literature. We stand with our original comment in that those discrepancies might be attributed to the cell type used. For instance, in our experiments, depletion of FERMT2 leads most strikingly to a change in the FA size (Fig. 4C) which is similar to what has been observed for podocytes but is different to what happens in fibroblasts. Changes in FA for all the knockdowns tested are in agreement with our transcriptomic data that show changes in adhesion genes for all the knockdowns.

5)What happens to VSVG traffic or RUSH-Cadherin traffic when cells are plated on Matrigel and Fibronectin? Reduction in secretory traffic of these is an important experiment which is missing to close the loop and validate the model presented. Authors must test these experiments either with cells grown on matrix alone or in combination with siRNA to SEC23B. Authors should also monitor ERES and transport carriers in this background.

Reply: We agree with the reviewer and intend to perform these experiments.

Update: We have included VSVG transport experiment in cells plated on plastic and Matrigel. The results show that there is indeed a decrease in the transport of VSVG when cells are plated on Matrigel compared to plastic substrate. This phenotype was further rescued after treatment with Focal Adhesion kinase inhibitor, confirming the role of FA and substrate nature on cargo transport. The data is show in in Figure 5E.

6)This is not such a major issue, but it would be good to see a comparison in SEC23A levels in siRNA knockdown condition in comparison to those when cells are grown on different substrates and in ROCK1, FEMRT2 knockdowns (blots of which authors already have in this manuscript).

Reply: We will assess the level of SEC23A at the protein level for cells plated on matrigel or Fibroblast-derived ECM.

Update: We have included the WB for SEC23A in cells grown on Matrigel (Fig. 5C and 5D). The results show that the protein levels decrease in agreement with qPCR measurements.

****Minor comments:****

1)Scale bars are missing in EM images in Figure 2H.

Reply: We will add the scales in our EM images

Revision Plan

Update: A scale bar was added (Fig. S1D)

2) Show molecular weight markers in Western blots in main figure 3E and supplementary figure S1E.

Reply: We will add molecular weight markers in our Western-Blots

Update: Molecular weight markers were added (Figures 3C and 5C)

Reviewer #2 (Significance (Required)):

I have looked at the manuscript from through the lens of a cell biologist as that is predominantly my area of expertise. In that respect I find the screen conducted by authors particularly interesting as they aim to connect how extracellular cues regulate the secretory pathway. A screen seems justified as there is no comprehensive understanding linking the two above-mentioned processes. Authors have done a functional interaction screen and analyzed a lot of images to identify candidates which either increase or decrease secretory traffic in combination with SEC23A and SEC23B. Such a functional screen has helped authors identify candidates which were otherwise missed in single siRNA knockdowns in their previous work from 2012. This definitely opens up interesting avenues to test the candidates identified in the screen in different physiological contexts and in disease as also the transcriptional program connecting Focal adhesion signaling with the regulation of components governing secretion. Such functional interaction screens could also be employed to identify crosstalk of different cellular processes with the regulation secretory pathway at ER as well as at the Golgi apparatus.

Reply: We thank reviewer #2 for the comments. As we mentioned in our reply to reviewer #1, we strongly believe that these results will encourage further research at the crossroads of membrane trafficking and mechanobiology.

Referees cross-commenting

I agree with the comments from both the referees that the manuscript is very interesting, most experiments are well controlled, but the quantification of focal adhesion phenotype in knockdowns need to be done in an extensive manner and secretion phenotypes need to be measured upon plating cells on different matrix to validate the model presented.

Reply: These two experiments will be included in our revision

Revision Plan

Update: As per the request of the reviewer, we have done the quantification of the focal adhesion phenotype upon siRNA knockdown of the functional interactors. We have also performed a VSVG assay of cells plated on plastic and Matrigel. The results are shown in Fig. 4C and Fig. 5E respectively.

Reviewer #3 (Evidence, reproducibility and clarity (Required)):

****Summary****

The authors use a synchronized cargo release assay following codepletion of either Sec23 paralog with cytoskeletal and associated proteins to identify potential functional interactions between COPII trafficking and the cytoskeleton. This screen yields a number of Sec23b functionally interacting molecules that stall cargo trafficking to various degrees within the secretory pathway upon codepletion, and in the case of MACF1 reduce ERES number despite not physically interacting. Depletion of the majority of the identified Sec23b functional interactors alone surprisingly caused the downregulation of Sec23a at the mRNA and protein levels, and cargo trafficking could be partially or fully rescued by Sec23a overexpression depending on the codepleted cytoskeletal factor. RNA-seq enrichment analysis and imaging of a focal adhesion marker suggest that genes involved in cell adhesion were differentially regulated following depletion of the cytoskeletal functional interactors. Finally, the authors show that Sec23a expression levels are reduced when cells are cultured on dishes with high amounts of ECM to induce focal adhesions, and that inhibition of focal adhesion kinase can rescue Sec23a expression levels.

****Major comments****

#1 The authors successfully implicate a group of cytoskeletal proteins and their actions at focal adhesions in negatively regulating Sec23a expression levels and COPII trafficking. This description of a shared, novel mode of COPII transcriptional regulation by cytoskeletal factors is convincingly shown to be at least a contributor to the delayed trafficking in the presence of focal adhesions. In general, the data are reproducible and use appropriate statistical analysis. However, a more robust description of the architecture of early secretory pathway would be beneficial, especially in the case of MACF1 codepletion which cannot be fully rescued by Sec23a-YFP overexpression. In contrast, trafficking during codepletion of FERMT2 is fully rescued by Sec23a-YFP despite both MACF1 and FERMT2 showing similar loss of Sec23a mRNA levels upon codepletion. This data suggests that while the trafficking delay in FERMT2 codepletion might be exclusively due to reduced Sec23a expression levels, there are likely additional causes for the trafficking delay observed in MACF1 codepletion.

Revision Plan

Reply: We thank the reviewer for the appreciation of our results and the importance they might bear for the field. The reviewer has very neatly highlighted that each of our interactor hits might have roles in the secretory pathway beyond the ER or independent of the expression levels of SEC23A. This phenomenon could also explain the differential rescue of the arrival of VSVG at the plasma membrane upon SEC23A overexpression in FERMT2 and MACF1 knockdowns (FIG 3F). For instance, MACF1 has been involved in Golgi to Plasma Membrane transport as well (Kakinuma et al. Exp. Cell Res. 2004, Burgo et al. Dev. Cell 2012). So a possibility is that SEC23A overexpression rescues only ER to Golgi transport but the lack of rescue in the compartment between Golgi and plasma membrane independent of SEC23A expression levels would result in reduced rescue in the case of MACF1 compared to FERMT2. To support this, in our revised manuscript, we will provide example images from the experiment.

Nonetheless, we agree that these are very important observations from Reviewer #3 and warrant a detailed discussion in the light of other interactors as well, which we intend to highlight in our revised manuscript.

Update: Since the further characterisation of the secretory pathway is not a major point that needs further experiments, we have not included this data anymore. The lack of rescue for MACF1 is appropriately discussed.

#2 While there is indeed a reduction in the number of ERESs following MACF1 codepletion, the authors report an even more dramatic reduction in 'transport intermediates / cell' as marked by COPI. However, as recent cryo-EM analysis of ERESs has definitively show, COPI exists stably at ERGIC membranes (1). Thus, an alternative possibility for the more dramatic reduction of COPI sites compared to Sec31a sites in Figures 2B-E is that ERGIC membranes are destabilized following MACF1 codepletion in a manner independent of Sec23a expression, and this destabilization compounds with reduced ERES number to ultimately delay trafficking. To more directly determine whether ERGIC membranes stability is regulated by MACF1, the authors should compare COPI and ERGIC-53 staining among MACF1 codepleted and FERMT2 codepleted cells with and without Sec23a-YFP overexpressed to levels that rescue cargo trafficking. If Sec23a-YFP restores the number of ERGIC punctae marked by these stains in FERMT2 but not MACF1 codepleted cells, it would suggest a role for MACF1 in forming or stabilizing ERGIC membranes which are known to associate with microtubules and WHAMM, an actin nucleator. Additionally, it would be useful to costain COPII with COPI or ERGIC-53 in control, MACF1 depleted, MACF1 codepleted, and MACF1 codepleted and Sec23a-YFP rescued cells to determine their colocalization. COPII and ERGIC membranes should be almost entirely coupled and juxtaposed in control cells and may be decoupled upon loss of MACF if plays a role in ERGIC membrane localization and stability. These proposed experiments are relevant because ERGIC membranes are sites of COPII cargo delivery and changes in ERGIC stability or localization would suggest an additional mechanism for cytoskeletal regulation of COPII trafficking. These immunofluorescence studies should be straightforward and completed in a few weeks.

Revision Plan

Reply: Although a possible additional role of MACF1 in the organisation of early secretory pathway, stability of ERES, etc., independent of the expression of SEC23A is interesting on its own, we believe that an extensive characterization of these possible roles/ pathways as proposed by the reviewer is beyond the scope this manuscript.

Update: This is now discussed in more detail.

#3 The choice to use VSVG and E-Cadherin for the synchronized release assays unfortunately convolutes interpreting the 'transport ratios' used by the authors to compare the effects of the various codepletions. Each protein progresses beyond the Golgi during secretion, and the authors choose to calculate the ratio of cargo intensity at the plasma membrane normalized to the total cellular cargo. This means that the synchronized release assays and calculated 'transport ratios' assay not only ER to Golgi trafficking, but also trafficking from the Golgi to the plasma membrane. In instances where Sec23a-YFP overexpression does not fully rescue the codepletion, it is possible that additional trafficking delays occur during Golgi to plasma membrane trafficking that cause the 'transport score' to decrease. Thus, the 'transport score' as the authors calculate it is needlessly nonspecific to COPII trafficking and should not be used to compare the codepletions for COPII functional interactors.

Reply: We agree that the “transport score” used here and in our previous genome-wide screen (Simpson et. al Nat. Cell Biol. 2012) does not allow us to distinguish between the individual transport substeps in the transport of VSVG from the ER to the plasma membrane. However, as we see in Fig 1E, the proteins that we have decided to follow in more detail in this study do have a clear ER transport block phenotype (except for CRKL). So for 6 out of 7 of these proteins, the images clearly show that the decrease in the “transport score” is due to a decreased ER to Golgi transport.

#4 To mitigate unwanted contributions of post-COPII trafficking events from altering 'transport scores,' the authors should use a cargo for synchronized release assays that does not progress past the Golgi such as α -Mannosidase II and quantify a ratio of the perinuclear cargo signal to whole cell signal. Ideally, the screen would be repeated with a more appropriate cargo generating new 'transport scores' for the full list of cytoskeletal proteins. However, this may not be feasible, and as such 'transport scores' based on a Golgi resident protein should at least be produced for the 7 Sec23b functional interactors featured in this manuscript. These Golgi 'transport scores' would add much needed quantification of ER to Golgi transport delays that currently can only be inferred from the representative images in Figure 1E, which unfortunately show significant heterogeneity among cells from the same image. The authors should also explicitly state that any 'transport score' from a synchronous release assay using a cargo destined for the plasma membrane will take into account trafficking rate changes due not only to COPII, but also COPI from the ERGIC to the Golgi, and transport carriers departing from the TGN. These synchronized release assays would likely take between a few weeks to a few months depending on their ability to automate image analysis.

Revision Plan

Reply: We consider that having a “Golgi transport score” won’t add any new information as the proteins that we have chosen to follow are the ones that show a strong ER-block phenotype. However, we agree that such a “Golgi score” would indeed be useful if one would like to study other interactors, for instance, the ones that induce transport acceleration.

Also, we don’t expect all cells to behave similarly as the level of knockdown might be slightly different or because of the cell to cell variability. Even in control conditions (no knockdown), this heterogeneity is evident. As suggested by the reviewer, in our revised manuscript we will explicitly state that a change in the transport scores could mean a change in any sub-step of the transport from the ER to the PM in our assay.

Update: This is now explicitly mentioned.

Minor comments

It would be useful for the authors to quantify the number of focal adhesions present from Vinculin stains from Figure 4C and 5C instead of just showing representative images. It would be interesting to determine if there is a meaningful relationship between focal adhesion number induced by the codepletions or tissue culture coating and Sec23a expression levels like in Figure 3D. Generally, the figures, text, and references were appropriate.

Reply: As also pointed out by the other reviewers we will quantify the FA changes

Update: As per the request of the reviewer, we have done the quantification of the focal adhesion phenotype upon siRNA knockdown of the functional interactors. The results are shown in Fig 4C.

Reviewer #3 (Significance (Required)):

In recent years, significant effort has been devoted to elucidating mechanisms by which COPII trafficking is modulated in response to cellular cues. These studies have revealed that changes in nutrient availability, growth factors, ER stress, autophagy, and T-cell activation all cause changes in COPII trafficking via unique gene expression, splicing, or post-translational control (2-7). This work elucidates a novel mechanism of transcriptional control driven by focal adhesions. Additionally, it provides a number of potentially useful Sec23a and Sec23b functional interactors among cytoskeletal factors for further study. These unexplored factors may have unique mechanism of COPII regulation that could contribute to our understanding ER export modulation. Altogether, this and similar works are building an increasingly complex set of regulatory pathways that when integrated ultimately dictate COPII trafficking kinetics.

The reported findings are not only relevant to those who study COPII trafficking, but also other fields where secretion is studied in the context of the ECM. This work would suggest that secretion of factors involved in crosstalk between cells, including in tumors, is likely to be controlled by the interactions of cells with ECM.

Reply: We thank reviewer #3 for the comments and insightful discussion about the limitations of our assay that we will highlight in the revised manuscript and in general for the insight into the early secretory pathway regulation. Furthermore, their explicit summary of how our study could bridge COPII trafficking, ECM signaling and the relevance to various pathophysiologies is highly appreciated.

Expertise keywords: cell biology, light microscopy, membrane trafficking

References

1. Weigel A V., Chang CL, Shtengel G, Xu CS, Hoffman DP, Freeman M, et al. ER-to-Golgi protein delivery through an interwoven, tubular network extending from ER. *Cell*. 2021 Apr;184(9):2412-2429.e16.

2. Farhan, H., Wendeler, M. W., Mitrovic, S., Fava, E., Silberberg, Y., Sharan, R., Zerial, M., & Hauri, H. P. (2010). MAPK signaling to the early secretory pathway revealed by kinase/phosphatase functional screening. *Journal of Cell Biology*, 189(6), 997-1011.

3. Zacharogianni, M., Kondylis, V., Tang, Y., Farhan, H., Xanthakis, D., Fuchs, F., Boutros, M., & Rabouille, C. (2011). ERK7 is a negative regulator of protein secretion in response to amino-acid starvation by modulating Sec16 membrane association. *EMBO Journal*, 30(18), 3684-3700.

4. Lillmann, K.D., V. Reiterer, F. Baschieri, J. Hoffmann, V. Millarte, M.A. Hauser, A. Mazza, N. Atias, D.F. Legler, R. Sharan, et al 2015. Regulation of Sec16 levels and dynamics links proliferation and secretion. *J. Cell Sci*. 128:670-682.

5. Liu, L., Cai, J., Wang, H., Liang, X., Zhou, Q., Ding, C., Zhu, Y., Fu, T., Guo, Q., Xu, Z., Xiao, L., Liu, J., Yin, Y., Fang, L., Xue, B., Wang, Y., Meng, Z. X., He, A., Li, J. L., ... Gan, Z. (2019). Coupling of COPII vesicle trafficking to nutrient availability by the IRE1 α -XBP1s axis. *Proceedings of the National Academy of Sciences of the United States of America*, 116(24), 11776-11785.

6. Jeong, Y.-T., Simoneschi, D., Keegan, S., Melville, D., Adler, N. S., Saraf, A., Florens, L., Washburn, M. P., Civasotto, C. N., Fenyő, D., Cuervo, A. M., Rossi, M., & Pagano, M. (2018). The ULK1-FBXW5-SEC23B nexus controls autophagy. *ELife*, 1-25.

7. Wilhelmi, I., Kanski, R., Neumann, A., Herdt, O., Hoff, F., Jacob, R., Preußner, M., & Heyd, F. (2016). Sec16 alternative splicing dynamically controls COPII

Revision Plan

transport efficiency. Nature Communications, 7, 12347.
<https://doi.org/10.1038/ncomms12347>

3. Description of the revisions that have already been incorporated in the transferred manuscript

4. Description of analyses that authors prefer not to carry out

Reviewer #3 suggested to robustly characterise the early secretory pathway, in response to the depletion of our interactors, for instance, the role of MACF1 in the organization and the stability of ERES. This view is also supported by reviewer #1. However, in our revised manuscript we would like to focus more on the novel aspect of our study (as highlighted by all the reviewers), namely how ECM signaling and changes in FAs affect SEC23A and possibly ER transport. For this, we would like to present a more quantitative outlook of the FA phenotype and concentrate on the transport experiments. The reason for not dwelling into a more extensive characterization of the early secretory pathway is that these experiments are very interesting on their own, and merit a separate study that would deconvolve in detail the individual trafficking steps, and their relation to SEC23A expression, ERES stability, and ECM signaling.

Reviewer #2 suggested that to better characterize the FA phenotype and solve the apparent discrepancies between our data and the literature, we could test FAK phosphorylation. As we mentioned in our reply to this point, we think that most of the discrepancies arise from the different cell types used. Nevertheless, we agree that a quantitative approach is needed for a better characterisation of FA phenotype, therefore we intend to perform quantification of the vinculin stainings.

June 3, 2022

RE: JCB Manuscript #202110081R

Dr. Juan Jung
European Molecular Biology Laboratory
Cell Biology & Biophysics
Meyerhofstrasse 1
Heidelberg 69117
Germany

Dear Dr. Jung:

Thank you for submitting your revised manuscript entitled "A high throughput SEC23 functional interaction screen reveals a role for focal adhesion and extracellular matrix signalling in the regulation of COPII subunit SEC23A". We would be happy to publish your paper in JCB pending final revisions necessary to meet our formatting guidelines (see details below). In your final revision, please be sure to address reviewer #2's final minor concerns with appropriate explanations and text edits.

A. MANUSCRIPT ORGANIZATION AND FORMATTING:

- 1) Text limits: Character count for Reports is < 20,000, not including spaces. Count includes abstract, introduction, combined results and discussion, and acknowledgments. Count does not include title page, figure legends, materials and methods, references, tables, or supplemental legends.
- 2) Figures limits: Reports may have up to 5 main text figures.
- 3) Figure formatting: Scale bars must be present on all microscopy images, including inset magnifications. Molecular weight or nucleic acid size markers must be included on all gel electrophoresis. Please ensure that Western blot exposures are representative of the quantification.
- 4) Statistical analysis: Error bars on graphic representations of numerical data must be clearly described in the figure legend. The number of independent data points (n) represented in a graph must be indicated in the legend. Statistical methods should be explained in full in the materials and methods. For figures presenting pooled data the statistical measure should be defined in the figure legends. Please also be sure to indicate the statistical tests used in each of your experiments (either in the figure legend itself or in a separate methods section) as well as the parameters of the test (for example, if you ran a t-test, please indicate if it was one- or two-sided, etc.). Also, if you used parametric tests, please indicate if the data distribution was tested for normality (and if so, how). If not, you must state something to the effect that "Data distribution was assumed to be normal but this was not formally tested."
- 5) Abstract and title: The abstract should be no longer than 160 words and should communicate the significance of the paper for a general audience. The title should be less than 100 characters including spaces. Make the title concise but accessible to a general readership.

* While your current title will be appreciated by the specialists, we do not feel that it will be accessible to a broader cell biology audience. Therefore, we suggest the following title: Regulation of the COPII secretory machinery via focal adhesions and extracellular matrix signaling *
- 6) Materials and methods: Should be comprehensive and not simply reference a previous publication for details on how an experiment was performed. Please provide full descriptions in the text for readers who may not have access to referenced manuscripts.
- 7) * Please be sure to provide the sequences for all of your primers/oligos and RNAi constructs in the materials and methods. You must also indicate in the methods the source, species, and catalog numbers (where appropriate) for all of your antibodies. Please also indicate the acquisition (e.g. film or digital imager model) and quantification methods for immunoblotting/western blots. *

8) Microscope image acquisition: The following information must be provided about the acquisition and processing of images:

- Make and model of microscope
- Type, magnification, and numerical aperture of the objective lenses
- Temperature
- Imaging medium
- Fluorochromes
- Camera make and model
- Acquisition software
- Any software used for image processing subsequent to data acquisition. Please include details and types of operations involved (e.g., type of deconvolution, 3D reconstitutions, surface or volume rendering, gamma adjustments, etc.).

10) Supplemental materials: There are strict limits on the allowable amount of supplemental data. Reports may have up to 3 supplemental figures. Please also note that tables, like figures, should be provided as individual, editable files. A summary of all supplemental material should appear at the end of the Materials and methods section.

13) ORCID IDs: ORCID IDs are unique identifiers allowing researchers to create a record of their various scholarly contributions in a single place. At resubmission of your final files, please consider providing an ORCID ID for as many contributing authors as possible.

Please note that JCB now requires authors to submit Source Data used to generate figures containing gels and Western blots with all revised manuscripts. This Source Data consists of fully uncropped and unprocessed images for each gel/blot displayed in the main and supplemental figures. Since your paper includes cropped gel and/or blot images, please be sure to provide one Source Data file for each figure that contains gels and/or blots along with your revised manuscript files. File names for Source Data figures should be alphanumeric without any spaces or special characters (i.e., SourceDataF#, where F# refers to the associated main figure number or SourceDataFS# for those associated with Supplementary figures). The lanes of the gels/blots should be labeled as they are in the associated figure, the place where cropping was applied should be marked (with a box), and molecular weight/size standards should be labeled wherever possible.

B. FINAL FILES:

****It is JCB policy that if requested, original data images must be made available to the editors. Failure to provide original images upon request will result in unavoidable delays in publication. Please ensure that you have access to all original data images prior to final submission.****

****The license to publish form must be signed before your manuscript can be sent to production. A link to the electronic license to publish form will be sent to the corresponding author only. Please take a moment to check your funder requirements before choosing the appropriate license.****

Thank you for this interesting contribution, we look forward to publishing your paper in Journal of Cell Biology.

Sincerely,

Elizabeth Miller, PhD
Monitoring Editor

Andrea L. Marat, PhD
Senior Scientific Editor

Journal of Cell Biology

Reviewer #1 (Comments to the Authors (Required)):

The authors have made adequate efforts to address prior concerns raised, and the study now appears to be ready for publication.

Reviewer #2 (Comments to the Authors (Required)):

I had previously looked at the manuscript by Jung et. al., from Review commons. As with the initial assessment the manuscript is very interesting due to the approach taken by the authors in designing their screen which allows them to identify functional interactors of Sec23A and Sec23B. Based on the identity of the interactors the authors put forth a model wherein the due to presence of more focal adhesions, there is a transcriptional repression of Sec23A which delays the secretory traffic from the ER. I have also looked at the rebuttal letter and the authors have done a very good job of addressing most of the concerns which were raised in the first round of revisions. The manuscript is strengthened, and the model put forth is validated in my opinion. The manuscript is ready for publication.

I have two comments on this version of the manuscript.

My biggest concern during the first round of revisions were regarding disparity in the results seen in the phenotypes upon downregulation of interactors identified in this screen and those in the published literature and hence the authors were asked to characterize the data better to quantify the focal adhesion phenotype. I am satisfied by the quantifications shown by the authors. I am also willing to agree to the argument put forth by these authors that the discrepancy in their phenotypes with those published in literature could be due to differences in cell lines used. In the light of this argument, this manuscript may have been strengthened if some of the experiments especially correlation in Sec23A levels, Focal adhesion numbers and size and secretory transport could have been demonstrated in another cell line like fibroblasts. This may be tedious due to issues with transfecting fibroblasts and the data presented in Fig 4 establishes the effects of downregulation of the interactors strongly on Focal adhesions in HeLa cells and must be given merit.

The second major concern was that authors had not checked transport defects in cargo by growing cells on Matrigel or Fibronectin, which was necessary to firmly establish their model. Authors have now responded to this by doing experiments wherein they grow cells on different concentrations of Matrigel and monitor levels of Sec32A protein and transport of VSVG. The data suggests that in cells showing high expression of VSVG, the effects are strong on a transport delay. I am curious and

slightly confused by the inverse correlation between Matrigel concentrations used and the Sec23A levels. Have the authors looked at Focal adhesions in these backgrounds and do they show a difference under different Matrigel concentrations as expected for their model (20 vs 4 and 0.4 μg). I understand that focus of the paper is not testing focal adhesions with varying Matrigel concentration and so I would recommend commenting on that in text especially if there is literature to support these differences. Secondly, the representative blot shown in Fig 5C is a bit difficult to interpret due to saturation of bands (at least visually). It is clear from the quantification the differences are not as strong as compared to Fig 3C in RNAi backgrounds. I recommend discussing this in text in line no 281. This might also explain why they only saw transport defects in high expressing cells upon Matrigel plating. Do authors have a better representative blot or can they adjust the levels for better representation. Minor point:

I may have missed this point in the first review but can the authors comment on the dosages of siRNA used in the screen where transport defects are seen only in double knockdown conditions compared to that in Fig 2D and Fig2E where in single siRNA treatment they see effects on ERES and transport intermediates. Because if the dosage was similar to that in the screen, it is important to highlight the fact that even though there is a decrease in both ERES and transport carriers (quite a large defect for Sec23B depletion), it does not necessarily translate to a cargo transport defect as measured by the authors.

Reviewer #3 (Comments to the Authors (Required)):

I went through the new manuscript and the point-by-point response of the authors to my initial comments. As far as I can see, the authors responded to all my initial comments. The new experiments with cells plated on different stiffness might not be state of the art in the field of mechanobiology, but then, this is a paper that focuses on secretion. Therefore, I think way how the authors dealt with it is absolutely fine.

As far as my cross-comments are concerned, the authors opted to not include a full characterization of the early secretory pathway. Although this would have been really nice, I think that it is not a "must have".

Dear Dr. Jung:

Thank you for submitting your revised manuscript entitled "A high throughput SEC23 functional interaction screen reveals a role for focal adhesion and extracellular matrix signalling in the regulation of COPII subunit SEC23A". We would be happy to publish your paper in JCB pending final revisions necessary to meet our formatting guidelines (see details below). In your final revision, please be sure to address reviewer #2's final minor concerns with appropriate explanations and text edits.

A. MANUSCRIPT ORGANIZATION AND FORMATTING:

Full guidelines are available on our Instructions for Authors page, <https://jcb.rupress.org/submission-guidelines#revised>. **Submission of a paper that does not conform to JCB guidelines will delay the acceptance of your manuscript.**

- 1) Text limits: Character count for Reports is < 20,000, not including spaces. Count includes abstract, introduction, combined results and discussion, and acknowledgments. Count does not include title page, figure legends, materials and methods, references, tables, or supplemental legends.
- 2) Figures limits: Reports may have up to 5 main text figures.
- 3) Figure formatting: Scale bars must be present on all microscopy images, including inset magnifications. Molecular weight or nucleic acid size markers must be included on all gel electrophoresis. Please ensure that Western blot exposures are representative of the quantification.
- 4) Statistical analysis: Error bars on graphic representations of numerical data must be clearly described in the figure legend. The number of independent data points (n) represented in a graph must be indicated in the legend. Statistical methods should be explained in full in the materials and methods. For figures presenting pooled data the statistical measure should be defined in the figure legends. Please also be sure to indicate the statistical tests used in each of your experiments (either in the figure legend itself or in a separate methods section) as well as the parameters of the test (for example, if you ran a t-test, please indicate if it was one- or two-sided, etc.). Also, if you used parametric tests, please indicate if the data distribution was tested for normality (and if so, how). If not, you must state something to the effect that "Data distribution was assumed to be normal but this was not formally tested."
- 5) Abstract and title: The abstract should be no longer than 160 words and should communicate the significance of the paper for a general audience. The title should be less than 100 characters including spaces. Make the title concise but accessible to a general readership.

We have shortened the abstract to fit into the 160 words limit

* While your current title will be appreciated by the specialists, we do not feel that it will be accessible to a broader cell biology audience. Therefore, we suggest the following title: Regulation of the COPII secretory machinery via focal adhesions and extracellular matrix signaling *

We find this new title appropriate, therefore we have changed it in the text

6) Materials and methods: Should be comprehensive and not simply reference a previous publication for details on how an experiment was performed. Please provide full descriptions in the text for readers who may not have access to referenced manuscripts.

7) * Please be sure to provide the sequences for all of your primers/oligos and RNAi constructs in the materials and methods. You must also indicate in the methods the source, species, and catalog numbers (where appropriate) for all of your antibodies. Please also indicate the acquisition (e.g. film or digital imager model) and quantification methods for immunoblotting/western blots. *

8) Microscope image acquisition: The following information must be provided about the acquisition and processing of images:

- a. Make and model of microscope
- b. Type, magnification, and numerical aperture of the objective lenses
- c. Temperature
- d. Imaging medium
- e. Fluorochromes
- f. Camera make and model
- g. Acquisition software
- h. Any software used for image processing subsequent to data acquisition. Please include details and types of operations involved (e.g., type of deconvolution, 3D reconstitutions, surface or volume rendering, gamma adjustments, etc.).

10) Supplemental materials: There are strict limits on the allowable amount of supplemental data. Reports may have up to 3 supplemental figures. Please also note that tables, like figures, should be provided as individual, editable files. A summary of all supplemental material should appear at the end of the Materials and methods section.

12) Conflict of interest statement: JCB requires inclusion of a statement in the acknowledgements regarding competing financial interests. If no competing financial interests exist, please include the following statement: "The authors declare no competing financial interests." If competing interests are declared, please follow your

statement of these competing interests with the following statement: "The authors declare no further competing financial interests."

13) ORCID IDs: ORCID IDs are unique identifiers allowing researchers to create a record of their various scholarly contributions in a single place. At resubmission of your final files, please consider providing an ORCID ID for as many contributing authors as possible.

Please note that JCB now requires authors to submit Source Data used to generate figures containing gels and Western blots with all revised manuscripts. This Source Data consists of fully uncropped and unprocessed images for each gel/blot displayed in the main and supplemental figures. Since your paper includes cropped gel and/or blot images, please be sure to provide one Source Data file for each figure that contains gels and/or blots along with your revised manuscript files. File names for Source Data figures should be alphanumeric without any spaces or special characters (i.e., SourceDataF#, where F# refers to the associated main figure number or SourceDataFS# for those associated with Supplementary figures). The lanes of the gels/blots should be labeled as they are in the associated figure, the place where cropping was applied should be marked (with a box), and molecular weight/size standards should be labeled wherever possible.

B. FINAL FILES:

-- Cover images: If you have any striking images related to this story, we would be happy to consider them for inclusion on the journal cover. Submitted images may also be chosen for highlighting on the journal table of contents or JCB homepage carousel. Images should be uploaded as TIFF or EPS files and must be at least 300

dpi resolution.

****It is JCB policy that if requested, original data images must be made available to the editors. Failure to provide original images upon request will result in unavoidable delays in publication. Please ensure that you have access to all original data images prior to final submission.****

****The license to publish form must be signed before your manuscript can be sent to production. A link to the electronic license to publish form will be sent to the corresponding author only. Please take a moment to check your funder requirements before choosing the appropriate license.****

Thank you for this interesting contribution, we look forward to publishing your paper in Journal of Cell Biology.

Sincerely,

Elizabeth Miller, PhD
Monitoring Editor

Andrea L. Marat, PhD
Senior Scientific Editor

Journal of Cell Biology

Reviewer #1 (Comments to the Authors (Required)):

The authors have made adequate efforts to address prior concerns raised, and the study now appears to be ready for publication.

Reviewer #2 (Comments to the Authors (Required)):

I had previously looked at the manuscript by Jung et. al., from Review commons. As with the initial assessment the manuscript is very interesting due to the approach taken by the authors in designing their screen which allows them to identify functional interactors of Sec23A and Sec23B. Based on the identity of the interactors the authors put forth a model wherein the due to presence of more focal adhesions, there is a transcriptional repression of Sec23A which delays the secretory traffic from the ER. I have also looked at the rebuttal letter and the authors have done a very good job of addressing most of the concerns which were raised in the first round of revisions. The manuscript is strengthened, and the model put forth is validated in my opinion. The manuscript is ready for publication.

I have two comments on this version of the manuscript.

My biggest concern during the first round of revisions were regarding disparity in the results seen in the phenotypes upon downregulation of interactors identified in this screen and those in the published literature and hence the authors were asked to characterize the data better to quantify the focal adhesion phenotype. I am satisfied by the quantifications shown by the authors. I am also willing to agree to the argument put forth by these authors that the discrepancy in their phenotypes with those published in literature could be due to differences in cell lines used. In the light of this argument, this manuscript may have been strengthened if some of the experiments especially correlation in Sec23A levels, Focal adhesion numbers and size and secretory transport could have been demonstrated in another cell line like fibroblasts. This may be tedious due to issues with transfecting fibroblasts and the data presented in Fig 4 establishes the effects of downregulation of the interactors strongly on Focal adhesions in HeLa cells and must be given merit.

The second major concern was that authors had not checked transport defects in cargo by growing cells on Matrigel or Fibronectin, which was necessary to firmly establish their model. Authors have now responded to this by doing experiments wherein they grow cells on different concentrations of Matrigel and monitor levels of Sec32A protein and transport of VSVG. The data suggests that in cells showing high expression of VSVG, the effects are strong on a transport delay.

I am curious and slightly confused by the inverse correlation between Matrigel concentrations used and the Sec23A levels. Have the authors looked at Focal adhesions in these backgrounds and do they show a difference under different Matrigel concentrations as expected for their model (20 vs 4 and 0.4 μ g). I understand that focus of the paper is not testing focal adhesions with varying Matrigel concentration and so I would recommend commenting on that in text especially if there is literature to support these differences.

We find statistically no differences between the different Matrigel concentrations. We have added this in the text and have modified the FIG5 to indicate the non-significance.

Secondly, the representative blot shown in Fig 5C is a bit difficult to interpret due to saturation of bands (at least visually). It is clear from the quantification the differences are not as strong as compared to Fig 3C in RNAi backgrounds. I

recommend discussing this in text in line no 281. This might also explain why they only saw transport defects in high expressing cells upon Matrigel plating. Do authors have a better representative blot or can they adjust the levels for better representation.

Indeed, the cells plated on Matrigel have relatively less downregulation of SEC23A compared to the cytoskeletal interactors (which is consistent with the mRNA levels) and hence only cells with higher cargo load show transport defects. We have mentioned this in the text.

We have made our quantifications in non-saturated blots. We have included these blots in the source material

Minor point:

I may have missed this point in the first review but can the authors comment on the dosages of siRNA used in the screen where transport defects are seen only in double knockdown conditions compared to that in Fig 2D and Fig2E where in single siRNA treatment they see effects on ERES and transport intermediates. Because if the dosage was similar to that in the screen, it is important to highlight the fact that even though there is a decrease in both ERES and transport carriers (quite a large defect for Sec23B depletion), it does not necessarily translate to a cargo transport defect as measured by the authors.

We have added an explanation for this in the text.

Reviewer #3 (Comments to the Authors (Required)):

I went through the new manuscript and the point-by-point response of the authors to my initial comments. As far as I can see, the authors responded to all my initial comments. The new experiments with cells plated on different stiffness might not be state of the art in the field of mechanobiology, but then, this is a paper that focuses on secretion. Therefore, I think way how the authors dealt with it is absolutely fine. As far as my cross-comments are concerned, the authors opted to not include a full characterization of the early secretory pathway. Although this would have been really nice, I think that it is not a "must have".